# Learning Erdős-Rényi Random Graphs
# via Edge Detecting Queries

**Zihan Li**
National University of Singapore
`lizihan@u.nus.edu`

**Matthias Fresacher**
University of Adelaide
`matthias.fresacher@adelaide.edu.au`

**Jonathan Scarlett**
National University of Singapore
`scarlett@comp.nus.edu.sg`

## Abstract

In this paper, we consider the problem of learning an unknown graph via queries on groups of nodes, with the result indicating whether or not at least one edge is present among those nodes. While learning arbitrary graphs with $n$ nodes and $k$ edges is known to be hard in the sense of requiring $\Omega(\min\{k^2 \log n, n^2\})$ tests (even when a small probability of error is allowed), we show that learning an Erdős-Rényi random graph with an average of $\bar{k}$ edges is much easier; namely, one can attain asymptotically vanishing error probability with only $O(\bar{k} \log n)$ tests. We establish such bounds for a variety of algorithms inspired by the group testing problem, with explicit constant factors indicating a near-optimal number of tests, and in some cases asymptotic optimality including constant factors. In addition, we present an alternative design that permits a near-optimal sublinear decoding time of $O(\bar{k} \log^2 \bar{k} + \bar{k} \log n)$.

## 1 Introduction

Graphs are a ubiquitous tool in modern statistics and machine learning for depicting interactions, relations, and physical connections in networks, such as social networks, biological networks, sensor networks, and so on. Often, the graph is not known *a priori*, and must be learned via queries to the network. In this paper, we consider the problem of graph learning via *edge detecting queries*, where each query contains a subset of the nodes, and the binary outcome indicates whether or not there is at least one edge among these nodes. See Section 1.1 for previous work on this problem.

An application of this problem highlighted in previous works such as [16] is that of learning which chemicals react with each other, using tests that are able to detect whether any reaction occurs. Another potential application is learning connectivity in large wireless networks: Each node has a unique identifier, and in response to a query, a node sends feedback to a central unit if the query includes both itself and at least one of its neighbors. Then, to attain the query outcome, the central unit only has to detect whether *any* feedback signal was received.

We consider the fundamental question of how many queries are needed to learn the graph. Under *adaptive testing* (i.e., tests can be designed based on previous outcomes), this question is well-understood [30], as outlined below. However, an impossibility result of [1] indicates that considerably more *non-adaptive* tests are needed in the worst-case sense for the class of graphs with a bounded number of edges. We show that this picture is much more positive in the *average-case sense* by studying the average performance with respect to Erdős-Rényi graphs [14]. In addition, to demonstrate that these findings are not overly reliant on the specific random graph model, we also present similar findings assuming only bounds on the number of edges and the maximum degree (see Appendix H).

## 1.1 Related Work

The problem considered in this paper can be viewed as a constrained group testing problem [8, Sec. 5.8]. We highlight the most relevant group testing works throughout the paper, and here simply refer the reader to [23] for a survey of the zero-error setting, and to [8] for a survey of the small-error setting (i.e., the algorithm is allowed a small probability of failure). These settings are fundamentally different, since the number of tests in the small error setting is $O(K \log N)$ (for $K$ defectives among $N$ items), while the zero-error criterion requires $\Omega(\min\{N, K^2\})$ tests.

Early works on graph learning via edge detecting queries considered identifying a single edge [3, 4] and then several edges [30] in a slightly more general scenario where the "defective graph" $G$ is known to be a sub-graph of a larger graph $H$. Several works considered specific graph classes such as matchings, stars, and cliques [9, 10, 27]. We particularly highlight the work of Johann [30], who gave an adaptive procedure requiring $k \log_2 \frac{|\widetilde{E}|}{k} + O(k)$ tests, where $\widetilde{E}$ is the set of edges in the larger graph $H$; this bound is optimal up to the $O(k)$ remainder term. More recently, extensions to hypergraphs have also been considered [2, 11, 12, 24].

While the adaptive setting is well-understood, the non-adaptive setting [1, 32] and adaptive settings with limited stages [1, 17, 28] are more challenging. We refer the reader to [1] for a recent survey of what is known, with a notable distinction between Monte Carlo and Las Vegas style algorithms. We highlight that in stark contrast with the standard group testing problem, the number of *non-adaptive* tests required to identify arbitrary graphs with $k$ edges and $n$ nodes is at least $\Omega(\min\{k^2 \log n, n^2\})$, *even under the small-error criterion.*[1]

## 1.2 Contributions

The $\Omega(\min\{k^2 \log n, n^2\})$ hardness result given in [1] holds with respect to worst-case graphs containing $k$ edges, which raises the question of whether some notion of *average-case* or further restricted graph classes can overcome this inherent difficulty. We focus primarily on the average case with respect to the ubiquitous Erdős-Rényi random graph model[2] and the small-error criterion, showing that indeed the number of tests required reduces to $O(\bar{k} \log n)$ for graphs with an average of $\bar{k}$ edges, and providing fairly tight explicit constant factors. In Appendix H, we describe how to attain similar results for general graphs with at most $k$ edges and maximum degree $d = o(\sqrt{k})$, albeit with slightly worse constant factors.

In more detail, we show the following for Erdős-Renyi random graphs:

- We provide a simple algorithm-independent lower bound based on counting the number of graphs within a high-probability set;
- We extend the COMP, DD, and SSS decoding algorithms [6, 19] from standard group testing to the graph learning problem, and provide upper and lower bounds on their asymptotic performance under a natural random test design.
- We propose a sublinear-time decoding algorithm (and its associated test design) based on the GROTESQUE algorithm [18], and show that it succeeds with high probability with $O(\bar{k} \log^2 \bar{k} + \bar{k} \log n)$ decoding time, thus nearly matching an $\Omega(\bar{k} \log \frac{n^2}{k})$ lower bound.

Briefly, the above-mentioned decoding algorithms are described as follows: COMP (*cf.*, Section 4.1) assumes all pairs are edges unless their nodes are both in some negative test, DD (*cf.*, Section 4.2) uses the COMP solution to identify "possible edges" and then declares a pair to be an edge only if it is the unique possible edge among the nodes in some test, and SSS (*cf.*, Section 5) solves an integer program to find the sparsest graph consistent with the test outcomes.

While the group testing algorithms themselves extend easily to our setting, their theoretical analyses require significant additional effort (see Appendix I for further discussion). For instance, for group testing, the analysis is symmetric with respect to any defective set of size $k$, whereas for graph learning, different graphs with a fixed number of edges can behave very differently, and even seemingly simple tasks (e.g., determining the probability of a positive test) become challenging.

With the exception of the sublinear-time decoding part, our results are summarized in Figure 1, where we plot the asymptotic ratio between the information-theoretic bound and the number of tests for each algorithm, for sparsity levels $\theta \in (0,1)$ such that $\bar{k} = \Theta(n^{2\theta})$. We observe the following:

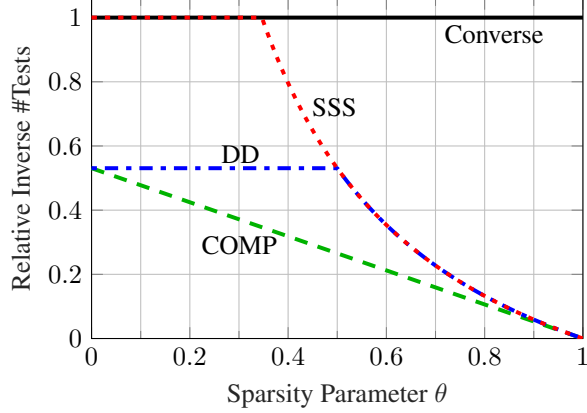

- For $\theta > \frac{1}{2}$, the DD upper bound and SSS lower bound match under i.i.d. random testing. As we explain in Section 5, SSS is the optimal algorithm, so if it fails then so does any algorithm. Hence, DD is asymptotically optimal (including constants) under i.i.d. random testing for $\theta > \frac{1}{2}$.

- For $\theta \leq \frac{1}{2}$, DD succeeds with fewer than twice as many tests as the optimal information-theoretic threshold; the latter is a converse bound applying to *any* test design (not only i.i.d. random testing).

Figure 1: Asymptotic values of $\frac{\bar{k}\log_2(1/q)}{\#\text{Tests}}$ for recovering Erdős-Rényi random graphs with edge probability $q = \Theta(n^{2(\theta-1)})$, and average number of edges $\bar{k} = q\binom{n}{2}$. The "COMP" and "DD" curves are achievability bounds, whereas the "Converse" and "SSS" curves are converse bounds for arbitrary test designs and i.i.d. random test designs, respectively).

While analogous results have been established for standard group testing [6], we again highlight that the analysis comes with several non-trivial challenges, particularly when it comes to DD and SSS. See Appendix I for an outline of the main differences.

## 2 Setup

We seek to learn an unknown undirected graph $G = (V, E)$ with $n$ nodes, i.e., the vertex set is $V = \{1, \ldots, n\}$, and the edge set $E$ contains up to $\binom{n}{2}$ pairs of nodes. We adopt a random graph model in which each edge appears in the graph independently with probability $q$ (i.e., the Erdős-Rényi graph $\text{ER}(n, q)$). After the graph $G$ is randomly drawn, it is fixed throughout the entire testing process (described below).

We test the nodes in groups; the output of each test takes the form

$$Y = \bigvee_{(i,j)\in E} \{X_i \cap X_j\}, \tag{1}$$

where the binary-valued test vector $X = (X_1, \ldots, X_n)$ indicates which nodes are included in the test. That is, the resulting output $Y = 1$ if and only if at least one edge exists in the sub-graph of $G$ induced by the nodes included in the test; we henceforth use the terminology that such an edge is *covered*. We refer to tests with $Y = 1$ as positive, and tests with $Y = 0$ as negative. A total of $t$ tests are performed according to the test vectors $X^{(1)}, \ldots, X^{(t)}$ to produce the outcomes $Y^{(1)}, \ldots, Y^{(t)}$. We focus on *non-adaptive* tests, where $X^{(1)}, \ldots, X^{(t)}$ must be selected prior to observing any outcomes.

Given the tests and their outcomes, a *decoder* forms an estimate $\widehat{G}$ of the graph $G$, or equivalently, an estimate $\widehat{E}$ of the edge set $E$. One wishes to design a sequence of tests $X^{(1)}, \ldots, X^{(t)}$, with $t$ ideally as small as possible, such that the decoder recovers $G$ with probability arbitrarily close to one. The error probability is given by

$$P_{\mathrm{e}} := \mathbb{P}[\widehat{G} \neq G], \tag{2}$$

and is taken over the randomness of the graph $G$, as well as the tests $X^{(1)}, \ldots, X^{(t)}$ (if randomized). We only consider deterministic decoding algorithms (without loss of optimality), and all of our results are asymptotic in the limit as $n \to \infty$ (with $q$ varying as a function of $n$).

## 2.1 Sparsity Level

We focus our attention on sparse graphs, i.e., $q = o(1)$ as $n \to \infty$.[3] More specifically, we consider the sublinear scaling regime $q = \Theta(n^{-2(1-\theta)})$ for some $\theta \in (0,1)$, meaning that the average number of edges $\bar{k} = \binom{n}{2}q$ behaves as $\Theta(n^{2\theta})$. By the assumption $\theta \in (0,1)$, we also have

$$n^{-(2-\eta)} \ll q \ll n^{-\eta}, \quad n^{\eta} \ll k \ll n^{2-\eta} \tag{3}$$

for sufficiently small (but constant) $\eta > 0$ and sufficiently large $n$. Here and subsequently, we write $f(n) \ll g(n)$ as a shorthand for $f(n) = o(g(n))$.

## 2.2 Bernoulli Random Testing

For the most part, we will focus on the case that the tests are designed randomly: Each node is independently placed in each test with a given probability $p$. We refer to this as *i.i.d. Bernoulli testing*, or simply *Bernoulli testing* for short. Analogous designs are known to lead to most of the best-known performance bounds in the group testing literature [6, 37], with the exception of some slight improvements shown recently via more structured random designs [20, 31].

We parametrize $p$ as $p = \sqrt{\frac{2\nu}{qn^2}}$ for some constant $\nu > 0$, as this scaling regime turns out to be optimal in all cases (with the choice $\nu = 1$ further being optimal for the algorithms we consider). Note that this choice of $p$ gives $p^2 = \frac{\nu}{k}(1 + o(1))$, since $\bar{k} = \frac{1}{2}qn^2(1 + o(1))$.

When studying probabilities associated with a single random Bernoulli test, we will denote the test outcome by $Y$, and the (random) indices of nodes included in the test by $\mathcal{L} \subseteq \{1, \ldots, n\}$. In addition, $\mathbb{P}_G[\cdot]$ denotes probability (with respect to the random testing alone) when the underlying graph is $G$.

## 2.3 Typical Graphs

Throughout the paper, we frequently make use of the following *typical set* of graphs:

$$\mathcal{T}(\epsilon_n) = \Big\{ G : (1 - \epsilon_n)\bar{k} \le k \le (1 + \epsilon_n)\bar{k}, d \le d_{\max}, $$
$$(1 - \epsilon_n)(1 - e^{-\nu}) \le \mathbb{P}_G[Y = 1] \le (1 + \epsilon_n)(1 - e^{-\nu}) \Big\}, \tag{4}$$

where $k = |E|$ is the number of edges, $d$ is the maximum degree of $G$, and

$$d_{\max} = \begin{cases} 2nq & \theta > \frac{1}{2} \\ \log n & \theta \le \frac{1}{2}. \end{cases} \tag{5}$$

The following lemma justifies the terminology *typical set* by showing that the random graph lies in this set with probability approaching one.

**Lemma 1.** *Fix $\theta \in (0,1)$, and let $G \sim \mathrm{ER}(n, q)$ for some $q = \Theta(n^{-2(1-\theta)})$. In addition, suppose that $\mathbb{P}_G[Y = 1]$ in (4) is defined with respect to $\mathrm{Bernoulli}(p)$ testing with $p = \sqrt{\frac{2\nu}{qn^2}}$ for fixed $\nu > 0$. Then, there exists a sequence $\epsilon_n \to 0$ such that $\mathbb{P}[G \in \mathcal{T}(\epsilon_n)] \to 1$ as $n \to \infty$.*

The condition on $k$ in the typical set simply states that the number of edges is close to its mean, which follows by standard concentration bounds. The bound on the maximum degree is similarly standard and straightforward to establish. By far the most challenging part is bounding $\mathbb{P}_G[Y = 1]$ with high probability; this is done using the inclusion-exclusion principle (i.e., Bonferroni's inequalities) and carefully bounding the probability of a random test containing one edge, two edges, three edges, and so on. The details are given in Appendix A.

Using the bounds on $k$ and $d$ in (4), along with the fact that $q$ satisfies (3), we readily observe that

$$d^2 \ll k, \quad dp \ll 1 \tag{6}$$

in both cases of (5). Note that the second of these statements follows immediately from the first since we focus on the regime $p = \Theta\big(\frac{1}{\sqrt{k}}\big)$.

**Algorithm 1** Combinatorial Orthogonal Matching Pursuit (COMP)
---
**Input:** Test designs $\{\mathcal{L}^{(i)}\}_{i=1}^{t}$, outcomes $\mathbf{Y} = (Y^{(1)}, \dots, Y^{(t)})$
  1: Initialize $\widehat{E}$ to contain all $\binom{n}{2}$ edges
  2: **for each** $i$ such that $Y^{(i)} = 0$ **do**
  3:      Remove all edges from $\widehat{E}$ whose nodes are both in $\mathcal{L}^{(i)}$
  4: **return** $\widehat{G} = (V, \widehat{E})$
---

## 3  Algorithm-Independent Converse Bound

To provide a benchmark for our upper bounds, we first provide a simple algorithm-independent lower bound on the number of tests for attaining asymptotically vanishing error probability, which is based on fairly standard counting arguments and Fano's inequality [21, Sec. 2.10].

**Theorem 1.** *Under the setup of Section 2 with $q = o(1)$ and an arbitrary non-adaptive test design, in order to achieve $P_{\mathrm{e}} \to 0$ as $t \to \infty$, it is necessary that*

$$t \geq \left( \bar{k} \log_2 \frac{1}{q} \right)(1 - \eta) \tag{7}$$

*for arbitrarily small $\eta > 0$.*

*Proof.* The proof is based on the fact that the prior uncertainty (entropy) is roughly $\binom{n}{2} q \log \frac{1}{q} = \bar{k} \log_2 \frac{1}{q}$ bits, whereas each test only reveals one bit of information. See Appendix B for details.  □

Using a similar analysis to [13], the preceding result can easily be strengthened to the *strong converse*, stating that $P_{\mathrm{e}}$ is not only bounded away from zero when $t$ is below the threshold given, but tends to one. On the other hand, the proof based on Fano's inequality extends more easily to noisy settings. Extending the result to adaptive test designs (e.g., again see [13]) is also straightforward, but in this paper we focus exclusively on non-adaptive designs.

## 4  Algorithmic Upper Bounds

### 4.1  COMP Algorithm

Adopting the terminology from the group testing literature, the COMP algorithm is described in Algorithm 1. The simple idea is that if two nodes appear in a negative test, then the corresponding edge must be absent from $G$. Hence, all such edges are ruled out, and the remaining edges are declared to be present. Once Lemma 1 is in place, the theoretical analysis of COMP becomes very simple and similar to standard group testing [19], leading to the following.

**Theorem 2.** *Under the setup of Section 2 with $q = \Theta(n^{2(\theta-1)})$ for some $\theta \in (0,1)$, and Bernoulli testing with parameter $\nu = 1$, the COMP algorithm achieves $P_{\mathrm{e}} \to 0$ as long as*

$$t \geq \left( 2e \cdot \bar{k} \log n \right)(1 + \eta) \tag{8}$$

*for arbitrarily small $\eta > 0$.*

*Proof.* The graph properties given in the definition (4) of $\mathcal{T}(\epsilon_n)$ facilitate a direct analysis of the probability that the two nodes of a given non-edge fail to be included together in any negative test, and a union bound over all non-edges establishes the claim. See Appendix C for details.  □

### 4.2  DD Algorithm

Since we work on the assumption that edges are rare (i.e., $q \ll 1$), one would expect that COMP's approach of assuming edges are present (unless immediately proven otherwise) can be highly suboptimal. The DD algorithm,[4] described in Algorithm 2, overcomes this limitation by assuming edges are absent unless immediately proven otherwise. The way to prove the presence of the edge

**Algorithm 2** Definite Defectives (DD)

---

**Input:** Test designs $\{\mathcal{L}^{(i)}\}_{i=1}^t$, outcomes $\mathbf{Y} = (Y^{(1)}, \ldots, Y^{(t)})$

1: Initialize $\widehat{E} = \emptyset$, and initialize PE to contain all $\binom{n}{2}$ edges
2: **for each** $i$ such that $Y^{(i)} = 0$ **do**
3:     Remove all edges from PE whose nodes are both in $\mathcal{L}^{(i)}$
4: **for each** $i$ such that $Y^{(i)} = 1$ **do**
5:     If the nodes from $\mathcal{L}^{(i)}$ cover exactly one edge in PE, add that edge to $\widehat{E}$
6: **return** $\widehat{G} = (V, \widehat{E})$

---

---

**Algorithm 3** Smallest Satisfying Set (SSS)

---

**Input:** Test designs $\{\mathcal{L}^{(i)}\}_{i=1}^t$, outcomes $\mathbf{Y} = (Y^{(1)}, \ldots, Y^{(t)})$

1: Find $\widehat{E}$ that minimizes $|\widehat{E}|$ subject to $\phi_{\widehat{E}}(\mathcal{L}^{(i)}) = Y^{(i)}$ for all $i = 1, \ldots, t$, where the function $\phi_E(\mathcal{L}) = \vee_{(i,j) \in E}\{\{i,j\} \subseteq \mathcal{L}\}$ corresponds to the observation model (1).
2: **return** $\widehat{G} = (V, \widehat{E})$

---

is to use COMP to rule out non-edges, mark the remaining pairs as *possible edges* (PE), and then look for positive tests containing only a single pair from PE. The analysis of DD is a fair bit more challenging than COMP, but gives an improved bound, as stated in the following.

**Theorem 3.** *Under the setup of Section 2 with $q = \Theta(n^{2(\theta-1)})$ for some $\theta \in (0, 1)$, and Bernoulli testing with parameter $\nu = 1$, the DD algorithm achieves $P_e \to 0$ as long as*

$$t \geq \big(2 \max\{\theta, 1-\theta\} e \cdot \bar{k} \log n\big)(1 + \eta) \tag{9}$$

*for arbitrarily small $\eta > 0$.*

*Proof.* The proof is based on analyzing the two steps separately. In the first step, we show that with high probability not too many non-edges are included in PE, and in the second step, we show that conditioned on this success event in the first step, each true edge is the unique PE in some test with high probability. The details are given in Appendix D, and the main differences to the standard group testing analysis [6, 40] are highlighted in Appendix I. $\qquad\square$

## 5 SSS Algorithm Lower Bound

It is a standard result that under any random graph model, the optimal decoder (in the sense of minimizing $P_e = \mathbb{P}[\widehat{G} \neq G]$) is the one that declares $\widehat{G}$ to be the most probable graph that would have produced the observation vector $\mathbf{Y} = (Y^{(1)}, \ldots, Y^{(t)})$ if it were the true graph. Under the Erdős-Rényi graph model, graphs with fewer edges are always more likely, so this decoder simply searches for the graph with the fewest edges that is *satisfying* in the sense of being consistent with $\mathbf{Y}$. This leads to the SSS algorithm described in Algorithm 3. Similarly to [34], this algorithm amounts to an integer program, which may be hard to solve efficiently in general.

Despite this computational challenge, a key utility of studying SSS is as follows. Since it is the optimal decoding algorithm, a lower bound on the number of tests it requires is also a lower bound for *any* decoding algorithm. In the following theorem, we provide such a lower bound with respect to random Bernoulli test designs. While such a lower bound is, in a sense, weaker than that of Theorem 1 (because that result holds for arbitrary test designs), it leads to the important conclusion that one cannot hope to improve on the bound for DD for $\theta > \frac{1}{2}$ unless one moves beyond Bernoulli test designs. See Figure 1 for an illustration.

**Theorem 4.** *Under the setup of Section 2 with $q = \Theta(n^{2(\theta-1)})$ for some $\theta \in (0, 1)$, and Bernoulli testing with an arbitrary choice of $\nu > 0$, the SSS algorithm yields $P_e \to 1$ whenever*

$$t \leq \big(2\theta e \cdot \bar{k} \log n\big)(1 - \eta) \tag{10}$$

*for arbitrarily small $\eta > 0$.*

---

**Algorithm 4** Group Testing Quick and Efficient (GROTESQUE) – Informal Outline

---

**Input:** Number of bundles $B$, inclusion probability $r$
 1: Form bundles $\mathcal{B}_1, \ldots, \mathcal{B}_B$ by independently including each node in each $\mathcal{B}_b$ with probability $r$
 2: Initialize $\widehat{E} = \emptyset$
 3: **for each** $b = 1, \ldots, B$ **do**
 4:     Perform a multiplicity test (*cf.*, Section 6.2) on $\mathcal{B}_b$
 5:     **if** multiplicity test returned "single edge" **then**
 6:         Perform location test (*cf.*, Section 6.3) on $\mathcal{B}_b$ and add the resulting edge to $\widehat{E}$
 7: **return** $\widehat{G} = (V, \widehat{E})$

---

*Proof.* The proof is based on the fact that if an edge is *masked* (i.e., its nodes never appear together in any test without those of a different edge), then removing that edge from $E$ will produce a smaller satisfying set, meaning that the algorithm fails to output $E$. The details are given in Appendix E, and the main differences to the standard group testing analysis [6] are highlighted in Appendix I. $\qquad\square$

## 6 Sublinear-Time Decoding

A standard implementation of COMP or DD yields decoding complexity $O(n^2 t)$, which may be infeasible when $n$ is large and decoding time is limited. To attain *sublinear-time* decoding, considerably different algorithms are needed, as one certainly cannot rely on marking non-edges one by one. In Algorithm 4, we informally outline a sublinear-time decoding algorithm that builds on the ideas of the GROTESQUE algorithm for group testing [18]. We find it most convenient to formally describe the key components while simultaneously performing the theoretical analysis; see Sections 6.2 and 6.3. For the purpose of understanding the algorithm, it suffices to note the following:

- A *multiplicity test* performs a number $t_{\mathrm{mul}}$ of group tests in which the items from a given bundle are included independently with probability $\frac{1}{\sqrt{2}}$. By counting the number of positive tests, one can determine with high probability whether or not the bundle covers exactly one edge.
- A *location test* performs a sequence of carefully-designed tests that permit the identification of the unique edge in a given bundle, provided that bundle indeed only covers one edge.

The resulting number of tests and runtime are given in the following theorem. Note that in contrast to the previous sections, here our focus is on the scaling laws and not the implied constants. This is due to the fact that attaining sharp constant factors with sublinear-time decoding has remained an open challenge even in the simpler group testing setting [15, 18, 29, 33].

**Theorem 5.** *Under the setup of Section 2 with $q = \Theta(n^{2(\theta-1)})$ for some $\theta \in (0, 1)$, the GROTESQUE test design and decoding algorithm achieves $P_{\mathrm{e}} \to 0$ with $t = O(\bar{k} \cdot \log \bar{k} \cdot \log^2 n)$ tests, and the decoding time behaves as $O(\bar{k} \log^2 \bar{k} + \bar{k} \log n)$ with probability approaching one.*

The proof is given below after a short discussion. While it may seem unusual to have a decoding time smaller than the number of tests, this is because the decoder is allowed to selectively decide which tests to make use of, and does not end up using them all. (We implicitly assume that fetching the result of a given test can be done in constant time.) Comparing to Theorems 2 and 3, we see that the number of tests performed has increased by a $\log \bar{k} \cdot \log n$ factor. On the other hand, the decoding time is nearly optimal: An analogous argument to Theorem 1 reveals an $\Omega(\bar{k} \log n)$ lower bound, and the upper bound in Theorem 5 matches this result when $\log^2 \bar{k} = O(\log n)$, and more generally comes within at most a single logarithmic factor.

We briefly mention that the *encoding* time (i.e., placing nodes in tests) is certainly not sublinear, so the advantage of sublinear decoding time is most beneficial when the encoding time does not pose a bottleneck (e.g., due to an efficient parallel implementation and/or pre-processing).

### 6.1 Proof Step 1 – Bundles of Tests

Since the number of edges $k$ and maximal degree $d$ behave as stated in (4) for some $\epsilon_n = o(1)$ with probability approaching one (see Lemma 1), it suffices to establish the claims of Theorem 5 conditioned on an arbitrary graph $G$ satisfying such properties. We implicitly condition on such a graph throughout the analysis.

As described in Algorithm 4, we form a number $B$ of "bundles" of tests, where each node is placed in each bundle with probability $r \in (0, 1)$. In Appendix F, we use a direct probabilistic analysis to show that under a choice satisfying $B = (4\bar{k} \log \bar{k})(1 + o(1))$, we have with probability $1 - o(1)$ that every edge is the unique one in at least one bundle.

## 6.2 Proof Step 2 – Multiplicity Tests

We perform a multiplicity test on each bundle by performing a series of (group) tests in which every node is independently included with probability $\frac{1}{\sqrt{2}}$. For each such test:

- If there are no edges, the output is always 0;
- If there is exactly one edge, each output equals 1 with probability $\frac{1}{2}$;
- If there are multiple edges, each output equals 1 with probability strictly higher than $\frac{1}{2}$. To see this, first observe that if there are two edges $e_1, e_2$ among 3 nodes, then the probability of a positive test is $\mathbb{P}[e_1 \cup e_2] = \mathbb{P}[e_1] + \mathbb{P}[e_2] - \mathbb{P}[e_1 \cup e_2] = \frac{1}{2} + \frac{1}{2} - \frac{1}{2\sqrt{2}} > 0.646$, whereas if there are two disjoint edges $e_1, e_2$ then a similar calculation yields $\mathbb{P}[e_1 \cup e_2] \geq \frac{3}{4}$. Hence, the overall probability of a positive test is at least $0.646$.

Based on these observations, we declare each bundle to have a single edge if and only if the proportion of 1's lies in $(0, 0.573)$. Trivially, if the number of edges is zero, we never make a mistake. On the other hand, if the number of edges is one or more than one, we can apply Hoeffding's inequality with a margin of at least $0.073$; hence, using $t_{\mathrm{mul}}$ tests we have $\mathbb{P}[\text{misclassification}] \leq 2e^{-2t_{\mathrm{mul}} \cdot 0.073^2}$. Taking the union bound over the $B$ bundles, and noting that $2 \times 0.073^2 > 0.01$, we find that we can classify all of the bundles correctly with probability approaching one when $t_{\mathrm{mul}} = (100 \log B)(1 + o(1))$, so that the total number of tests used is $t_{\mathrm{mul}} B = (100 B \log B)(1 + o(1))$.

## 6.3 Proof Step 3 – Location Tests

For location testing, we assign each node a unique binary string of length $L = \lceil \log_2 n \rceil$. In the following, we consider an arbitrary bundle containing a single edge. For ease of presentation, we describe the location test as though the algorithm could adaptively perform tests, and then we describe how the same can be done non-adaptively.

**Adaptive location test.** The following procedure constructs two binary strings $A$ and $B$ of length $L$; these strings will index the two nodes in the bundle that have an edge between them. For each $\ell = 1, \ldots, L$, we do the following:

1. Test all nodes with a 0 in their $\ell$-th bit. If the test is positive, label both $A_\ell$ and $B_\ell$ as zero.
2. Test all nodes with a 1 in their $\ell$-th bit. If the test is positive, label both $A_\ell$ and $B_\ell$ as one.
3. If neither of the preceding tests is positive, we know that $A_\ell \neq B_\ell$, so we do the following:
   (a) If this is the first $\ell$ for which this case is encountered, then assign $A_\ell = 0$ and $B_\ell = 1$ (the other way around would be equally valid).
   (b) Otherwise, let $\ell' < \ell$ be an index where we encountered this case, and do the following:
      i. Let $v \in \{0, 1\}$ be the bit value that was assigned to $A_{\ell'}$.
      ii. Perform a test containing all nodes whose bit strings $(v_1, \ldots, v_L)$ have $v_\ell = v_{\ell'} = v$ and also all those that have $v_\ell = v_{\ell'} = 1 - v$.
      iii. If the test is positive, then assign $A_\ell = v$ and $B_\ell = 1 - v$. Otherwise, assign $A_\ell = 1 - v$ and $B_\ell = v$.

The idea of step 3(b)(ii) is that we already know that the edges corresponding to $A$ and $B$ must have $v$ and $1 - v$ in bit position $\ell'$ respectively, so we perform the test described to check whether the same is true of position $\ell$. If it is not true, then with $A$ and $B$ differing in their $\ell$-th bit, the only remaining case is that $A$ and $B$ have $1 - v$ and $v$ in bit position $\ell$ respectively.

**Non-adaptive location test.** The only types of tests that the adaptive algorithm above uses are (i) those used in Steps 1 and 2 with all nodes having a given $\ell$-th bit value; and (ii) those used in Step 3b containing all nodes with some $v_\ell = v_{\ell'} = v$ and some other $v_\ell = v_{\ell'} = 1 - v$. There are only $2L$ possible such tests of type (i), and $2\binom{L}{2}$ possible tests of type (ii). Hence, we can perform the tests non-adaptively by taking all such possible tests in advance. Moreover, we don't have to look at all their outcomes, but rather only those that we would have taken in the adaptive setting.

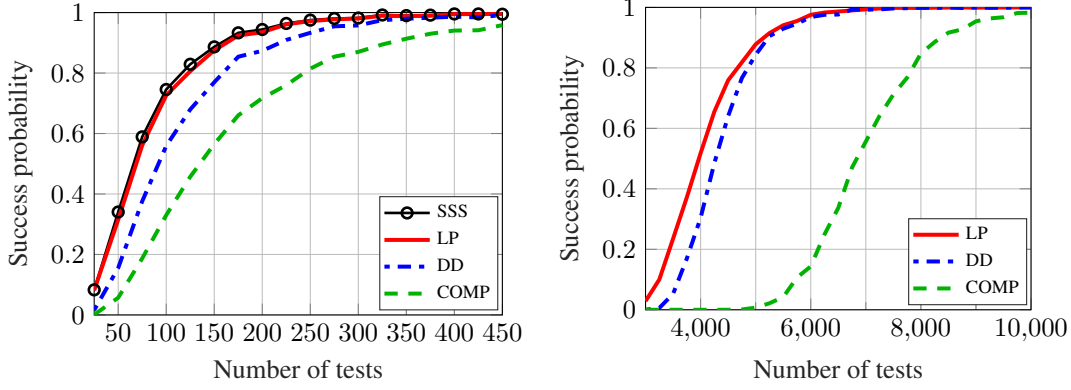

Figure 2: Performance of the COMP, DD, LP, and SSS algorithms for noiseless group testing under Bernoulli testing with $\nu = 1$, and with $n = 50$ and $\bar{k} = 5$ (Left); $n = 200$ and $\bar{k} = 200$ (Right).

Hence, the number of group tests per location test is at most $2\lceil \log_2 n \rceil + 2 \cdot \frac{1}{2} \lceil \log_2 n \rceil^2 = (\log_2 n)^2 (1 + o(1))$, so if we perform one location test for each bundle (and again only actually use those that we need) then the total is $B(\log_2 n)^2 (1 + o(1))$.

### 6.4 Proof Step 4 – Total Number of Tests and Decoding Time

The claims on the number of tests and runtime stated in Theorem 5 follow easily from the above analysis, and the details are deferred to Appendix F.

## 7 Numerical Experiments

We complement our theoretical findings with numerical experiments comparing COMP, DD, SSS, and a linear programming (LP) relaxation of SSS (analogous to [34]).[5] Figure 2 shows the success probability as a function of the number of tests in two cases: (i) $n = 50$ and $\bar{k} = 5$; (ii) $n = 200$ and $\bar{k} = 200$. In each case, we set $\nu = 1$ and compute the error probability averaged over 2000 trials. In the first case, we observe that the SSS and LP curves are very close, and require the fewest tests; DD requires more tests, and COMP requires the most. In the second case, we omit SSS due to its computational complexity, but we observe a similar ordering between LP, DD, and COMP. In both cases, the relative performance between the algorithms is consistent with our theoretical findings:

- The first case is a sparse setting, and the performance curves for COMP and DD are relatively closer, which is consistent with the fact that COMP and DD achieve the same theoretical bound in the sparse limit $\theta \to 0$ (see Figure 1).
- The second case is a denser setting, and the gap between LP and DD is narrower, which is consistent with the fact that the theoretical bounds for DD and SSS coincide in denser regimes.

In Appendix G, we provide similar plots for varying choices of $(\bar{k}, n)$ in order to demonstrate that the dependence of the number of tests on $\bar{k}$ and $n$ is in general agreement with our theory.

## 8 Conclusion

We have studied the problem of learning Erdős-Rényi random graphs via edge detecting queries, and demonstrated significantly improved scaling of $O(\bar{k} \log n)$ compared to worst-case graphs with $k$ edges. We provided order-optimal bounds for the COMP, DD, and SSS algorithms with explicit constants, showed DD to be optimal under Bernoulli testing when the graph is sufficiently dense ($\theta \geq \frac{1}{2}$), and introduced a sublinear-time algorithm that succeeds with $O(\bar{k} \log^2 \bar{k} + \bar{k} \log n)$ runtime.

Given that the ideas of this paper build on a variety of techniques for small-error group testing, it is natural to pursue further research in directions that were done previously in that setting, including separate decoding of items (edges) [35, 38], information-theoretic achievability bounds [20, 37], and near-constant tests-per-item (tests-per-node) designs [20, 31]. Generalizations of our techniques to hypergraph learning [2, 11, 12, 24] would also be of significant interest.

**Acknowledgment.** This work was supported by an NUS Early Career Research Award.

## Footnotes

[1]Note that the number of items $N$ in the standard group testing corresponds to $\binom{n}{2} = \Theta(n^2)$ in the graph learning problem with $n$ nodes, since *pairs of nodes* (i.e., potential edges) play the role of items. See Appendix I for a brief description of the group testing problem.

[2]More precisely, we consider the variant introduced by Gilbert [26].

[3]In fact, the arguments given in [5] can be applied to the present setting to show that one essentially cannot improve on individual testing of edges when $q = \Theta(1)$.

[4] For the graph learning problem, one may prefer to name the algorithm *Definite Edges*, but we prefer to maintain consistency with the group testing literature [6].

[5]The code is available at `https://github.com/scarlett-nus/er_edge_det`.

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
