[Supplementary Material]

# Supplementary Material

## Learning Erdős-Rényi Random Graphs via Edge Detecting Queries

### Zihan Li, Matthias Fresacher, and Jonathan Scarlett (NeurIPS 2019)

All citations below are to the reference list in the main body.

## A  Proof of Lemma 1 (High Probability Typicality)

To prove that there exists a sequence $\epsilon_n \to 0$ such that $\mathbb{P}[G \in \mathcal{T}(\epsilon_n)] \to 1$ as $n \to \infty$, it suffices to show that $\mathbb{P}[G \in \mathcal{T}(\epsilon)] \to 1$ as $n \to \infty$ for arbitrarily small (but fixed) $\epsilon > 0$.

Since $q = \Theta(n^{-2(1-\theta)})$ for some $\theta \in (0,1)$, the average number of edges $\bar{k} = q\binom{n}{2}$ must grow unbounded as $n \to \infty$. Hence, the fact that $\mathbb{P}[(1-\epsilon)\bar{k} \le k \le (1+\epsilon)\bar{k}] \to 1$ follows via basic binomial concentration (e.g., Chernoff bound). The bound on the degree is also easy to handle:

- For $\theta > \frac{1}{2}$, the per-node degree follows a binomial distribution with mean $(n-1)q = \Theta(n^c)$ for some $c > 0$. Hence, by the multiplicative form of the Chernoff bound [36, Sec. 4.1], the probability of the degree exceeding $2nq$ (which exceeds double the average) behaves as $e^{-\Omega(n^c)}$. By a union bound over the $n$ nodes, we find that the probability of any node's degree exceeding $2nq$ is at most $ne^{-\Omega(n^c)} \to 0$.
- For $\theta \le \frac{1}{2}$, it suffices to consider the case $\theta = \frac{1}{2}$, for which the probability of the maximal degree exceeding $\log n$ is clearly highest. In this case, a given node's degree follows a binomial distribution with $n - 1$ trials and success probability $\Theta(\frac{1}{n})$, so the mean behaves as $\Theta(1)$. Therefore, the probability of the degree exceeding $\log n$ equals the probability of being at least a factor $1 + \Delta$ higher than the mean $(n-1)q$, for some $\Delta = \Theta(\log n)$. By a standard form of the Chernoff bound [36, Sec. 4.1], this occurs with probability at most $e^{-(n-1)q[(1+\Delta)\log(1+\Delta)-\Delta]}$, which behaves as $e^{-\Omega(\log n \cdot \log \log n)}$ since $(n-1)q = \Theta(1)$. By a union bound over the $n$ nodes, the probability of any degree exceeding $\log n$ is at most $ne^{-\Omega(\log n \cdot \log \log n)} \to 0$.

By far the most challenging event to handle in the typical set is the final one, $(1-\epsilon)(1-e^{-\nu}) \le \mathbb{P}_G[Y=1] \le (1+\epsilon)(1-e^{-\nu})$. The intuition behind the analysis is as follows:

- In generic notation, let $A_1, \dots, A_N$ be independent events each occurring with probability $r$. Then $\mathbb{P}\big[\bigcup_{i=1}^N A_i\big] = 1 - (1-r)^N$, which behaves as $(1 - e^{-Nr})(1 + o(1))$ as $N \to \infty$ with $r = \Theta(\frac{1}{N})$.
- Letting $S_j = \sum_{1 \le i_1 < \dots < i_j \le N} \mathbb{P}[A_{i_1} \cap \dots \cap A_{i_j}]$, we know from the Bonferroni inequalities [25] that

$$\mathbb{P}\Big[\bigcup_{i=1}^N A_i\Big] \le \sum_{j=1}^{j_{\max}} (-1)^{j+1} S_j \tag{11}$$

  for odd $j_{\max}$, and the inequality is flipped for even $j_{\max}$.
- In the special case of independent events and $N \to \infty$ with $r = \Theta(\frac{1}{N})$, assuming that $j_{\max} = O(1)$, we have

$$\sum_{j=1}^{j_{\max}} (-1)^{j+1} S_j = \sum_{j=1}^{j_{\max}} (-1)^{j+1} \binom{N}{j} r^j \tag{12}$$

$$= -\sum_{j=1}^{j_{\max}} \frac{1}{j!} (-Nr)^j (1 + o(1)), \tag{13}$$

  since $\binom{N}{j} = \frac{N!}{j!(N-j)!} = \frac{1}{j!} N^j (1 + o(1))$. Due to the limit $\sum_{j=1}^{\infty} \frac{1}{j!}(-Nr)^j = e^{-Nr} - 1$, (13) is arbitrarily close to $1 - e^{-Nr}$ for large $j_{\max}$, regardless of whether $j_{\max}$ is even or odd. As one should expect, this matches the probability computed directly in the first dot point above.
- For the graph learning problem, we do not have independent events, but we will still show similar behavior to (13) to deduce the precise probability of a positive test given $G$.

We now proceed with the formal argument.

**High-probability counting event.** Fix an arbitrary integer $j_{\max} > 0$, and for each $j = 1, \ldots, j_{\max}$ and $\ell = 1, \ldots, 2j - 1$, let $U_{j,\ell}(G)$ be the number of sets of exactly $j$ edges in $G$ that collectively consist of exactly $\ell$ nodes.[6] We define the following typicality event for a graph $G$:

$$U_{j,\ell}(G) \leq \frac{1}{\delta} \binom{n}{\ell} \binom{\binom{\ell}{2}}{j} q^j, \quad \forall j = 1, \ldots, j_{\max}, \quad \ell = 1, \ldots, 2j - 1. \tag{14}$$

We claim that the probability (with respect to $G \sim \mathrm{ER}(n, q)$) of this occurring is at least $1 - O(\delta)$, where the implied constant depends on $j_{\max}$.

To see this, first note that any fixed collection of $j$ node pairs are all edges with probability $q^j$. The number of such collections satisfying the condition defining $U_{j,\ell}(G)$ is at most $\binom{n}{\ell}\binom{\binom{\ell}{2}}{j}$, i.e., first choose $\ell$ nodes from $n$, and then choose $j$ edges from the corresponding $\binom{\ell}{2}$ possibilities. Hence,

$$\mathbb{E}[U_{j,l}(G)] \leq \binom{n}{\ell} \binom{\binom{\ell}{2}}{j} q^j, \tag{15}$$

and the $1 - O(\delta)$ probability claim follows from Markov's inequality, a union bound over $j = 1, \ldots, j_{\max}$ and $\ell = 1, \ldots, 2j - 1$, and the assumption that $j_{\max}$ is finite.

**Analysis of $\mathbb{P}_G[Y = 1]$ with respect to a random test.** Let $G = (V, E)$ be a fixed graph whose number of edges $k$ and maximal degree $d$ satisfies the typicality bounds in (4), and that also satisfies the high-probability counting event (14). We write

$$\mathbb{P}[Y = 1] = \mathbb{P}\left[ \bigcup_{e \in E} A_e \right], \tag{16}$$

where $A_e$ is the event that both nodes from $e$ are in the test (and hence $\mathbb{P}[A_e] = p^2$), and here and subsequently we implicitly condition on $G$ being the graph (i.e., we write $\mathbb{P}[\cdot]$ in place of $\mathbb{P}_G[\cdot]$). The Bonferroni inequality therefore states that

$$\mathbb{P}[Y = 1] \leq \sum_{j=1}^{j_{\max}} (-1)^{j+1} S_j \tag{17}$$

for odd $j$ (or the reverse for even $j$), where

$$S_j = \sum_{1 \leq i_1 < \ldots < i_j \leq k} \mathbb{P}\left[ A_{e(i_1)} \cap \ldots \cap A_{e(i_j)} \right], \tag{18}$$

and where $A_{e(i)}$ is the $i$-th edge for some arbitrary but fixed ordering of the $k$ edges.

We proceed by characterizing $S_j$ for fixed $j$, assuming $j_{\max} = O(1)$ throughout. We will show that the summation in (18) is asymptotically equivalent to the restricted summation in which the $j$ edges are disjoint, i.e., share no nodes in common. First note that for disjoint $e(i_1), \ldots, e(i_j)$, we have $\mathbb{P}\left[ A_{e(i_1)} \cap \ldots \cap A_{e(i_j)} \right] = p^{2j}$. Since there are trivially at most $\binom{k}{j}$ ways of choosing $j$ disjoint edges, it follows that

$$\sum_{\substack{1 \leq i_1 < \ldots < i_j \leq k \\ \text{edges disjoint}}} \mathbb{P}\left[ A_{e(i_1)} \cap \ldots \cap A_{e(i_j)} \right] \leq \binom{k}{j} \cdot p^{2j} \tag{19}$$

$$= \left( \frac{1}{j!} \cdot k^j \cdot p^{2j} \right)(1 + o(1)), \tag{20}$$

since $\frac{k!}{(k-j)!} = k^j(1 + o(1))$ as $k \to \infty$ with $j = O(1)$.

We now seek a matching lower bound to (20). The number of unordered sequences of $j$ disjoint edges is equal to $\frac{1}{j!}$ times the number ordered sequences of $j$ disjoint edges, and to count the

latter, we consider a sequential selection of nodes. Whenever a node is selected, at most $d+1$ nodes are ruled out due to being its neighbor (or itself), so the number of selections is at least $k(k-(d+1))(k-2(d+1))\ldots(k-(j-1)(d+1))$. But since $j=O(1)$ and $d=o(k)$ (see (6)), this simply behaves as $k^j(1+o(1))$, and we deduce that

$$\sum_{\substack{1\leq i_1<\ldots<i_j\leq k \\ \text{edges disjoint}}} \mathbb{P}\big[A_{e(i_1)}\cap\ldots\cap A_{e(i_j)}\big] \geq \Big(\frac{1}{j!}\cdot k^j\cdot p^{2j}\Big)(1+o(1)). \tag{21}$$

We now show how to use (20) and (21) to deduce upper and lower bounds on $S_j$. In fact, the lower bound is trivial, since we can simply drop any remaining terms (i.e., those with non-disjoint edges) in (18) by lower bounding the summand by zero. For the upper bound, however, some additional effort is required. We let $N(i_1,\ldots,i_j)$ denote the number of nodes that the edges $e(i_1),\ldots,e(i_j)$ collectively contain, and write

$$S_j = \sum_{\ell=1}^{2j} \sum_{\substack{1\leq i_1<\ldots<i_j\leq k \\ N(i_1,\ldots,i_k)=\ell}} \mathbb{P}\big[A_{e(i_1)}\cap\ldots\cap A_{e(i_j)}\big]. \tag{22}$$

We bound the inner summand separately for each $\ell$. Note that $\ell=2j$ corresponds to the disjoint case that we already handled, so we proceed by assuming that $\ell<2j$.

For each $\ell<2j$, the summand in (22) is equal to $p^\ell$, and according to the high-probability event in (14), the number of such summands is at most $U_{j,\ell}\leq\frac{1}{\delta}\binom{n}{\ell}\binom{\binom{\ell}{2}}{j}q^j$, which we can crudely further upper bound as

$$U_{j,\ell} \leq O\Big(\frac{1}{\delta}\Big)\cdot n^\ell q^j, \tag{23}$$

since $j$ and $\ell$ are both $O(1)$. Since $\ell<2j$, or equivalently $\ell\leq 2j-1$ or $j\geq\frac{\ell+1}{2}$, we can write $q^j\leq q^{\ell/2}\cdot q^{1/2}$ (note that $q<1$), which implies

$$U_{j,\ell} \leq O\Big(\frac{1}{\delta}\Big)\cdot n^\ell q^{\ell/2}q^{1/2}, \tag{24}$$

and hence

$$\sum_{\substack{1\leq i_1<\ldots<i_j\leq k \\ N(i_1,\ldots,i_k)=\ell}} \mathbb{P}\big[A_{e(i_1)}\cap\ldots\cap A_{e(i_j)}\big] \leq O\Big(\frac{1}{\delta}\Big)\cdot n^\ell q^{\ell/2}q^{1/2}p^\ell \tag{25}$$

$$\leq O\Big(\frac{q^{1/2}}{\delta}\Big)\cdot k^{\ell/2}p^\ell, \tag{26}$$

where we have used $k=\big(\frac{1}{2}qn^2\big)(1+o(1))$. Now observe that if we choose $\delta$ to tend to zero at a strictly slower rate than $q^{1/2}$, the $O(\cdot)$ term in (26) behaves as $o(1)$. Since $kp^2=\Theta(1)$ by our choice of $p$, and $\ell=O(1)$ by assumption, we conclude that overall (26) vanishes as $n\to\infty$, for any $\ell<2j$. In contrast, the term (21) corresponding to $\ell=2j$ behaves as $\Theta(1)$. Since there are only a finite number of $\ell$ values, we deduce that the overall outer sum (22) is asymptotically equivalent to its first term characterized in (20)–(21):

$$S_j = \Big(\frac{1}{j!}\cdot k^j\cdot p^{2j}\Big)(1+o(1)). \tag{27}$$

By identifying this asymptotic expression with (13), with $N=k$ and $r=p^2$, we conclude that the following holds as $j_{\max}\to\infty$:

$$\mathbb{P}[Y=1] = \big(1-e^{-kp^2}\big)(1+o(1)) = (1-e^{-\nu})\cdot(1+o(1)), \tag{28}$$

since $p^2=\frac{\nu}{k}=\frac{\nu}{k}(1+o(1))$. In other words, by choosing $j_{\max}$ sufficiently large, we can ensure that under the high-probability event in (14) and the high-probability bounds on $k$ and $d$ in (4), it holds that $(1-e^{-\nu})(1-\epsilon)\leq\mathbb{P}[Y=1]\leq(1-e^{-\nu})(1+\epsilon)$ for arbitrarily small $\epsilon>0$ and sufficiently large $n$. This completes the proof of Lemma 1.

# B Proof of Theorem 1 (Converse Bound)

We use a conditional form of Fano's inequality (e.g., [39, Thm. 3]) with conditioning on the event that the number of edges $k$ in $G$ satisfies $(1 - \epsilon)\bar{k} \le k \le \bar{k}(1 + \epsilon)$ for small $\epsilon > 0$. Denoting this event by $\mathcal{A}$, and using the usual notation $H(X)$, $H(Y|X)$, $I(X;Y)$, etc. for entropy and mutual information, Fano's inequality gives

$$P_{\mathrm{e}} \ge \mathbb{P}[\mathcal{A}]\frac{H(G|\widehat{G}, \mathcal{A} = \mathrm{true}) - \log 2}{\log|\mathcal{G}_{\mathcal{A}}|} \tag{29}$$

$$= \mathbb{P}[\mathcal{A}]\frac{H(G|\mathcal{A} = \mathrm{true}) - I(G;\widehat{G}|\mathcal{A} = \mathrm{true}) - \log 2}{\log|\mathcal{G}_{\mathcal{A}}|}, \tag{30}$$

where $\mathcal{G}_{\mathcal{A}}$ is the set of graphs such that $(1 - \epsilon)\bar{k} \le k \le \bar{k}(1 + \epsilon)$.

Note that the preceding condition on $k$ is a standard notion of *typicality* for collections of independent random variables (in this case, edges). Using standard properties of typical sets [41, App. C], we have $\mathbb{P}[\mathcal{A}] = 1 - o(1)$, $\log|\mathcal{G}_{\mathcal{A}}| = \binom{n}{2}H_2(q)(1 + o(1))$, and $H(G|\mathcal{A} = \mathrm{true}) = \binom{n}{2}H_2(q)(1 + o(1))$, where $H_2(q) = q\log\frac{1}{q} + (1 - q)\log\frac{1}{1-q}$ is the binary entropy function. In addition, the data processing inequality [21, Sec. 2.8] gives $I(G;\widehat{G}|\mathcal{A} = \mathrm{true}) \le I(G;\mathbf{Y}|\mathcal{A} = \mathrm{true})$, and since $\mathbf{Y} \in \{0,1\}^t$, this mutual information is further upper bounded by $t\log 2$. Substituting the preceding findings into (30) yields

$$P_{\mathrm{e}} \ge \left(1 - \frac{t\log 2}{\binom{n}{2}H_2(q)}\right)(1 + o(1)). \tag{31}$$

Since we consider the regime $q \to 0$, we have $H_2(q) = \left(q\log\frac{1}{q}\right)(1 + o(1))$, and hence

$$P_{\mathrm{e}} \ge \left(1 - \frac{t\log 2}{\frac{1}{2}qn^2\log\frac{1}{q}}\right)(1 + o(1)). \tag{32}$$

Since $\bar{k} = \frac{1}{2}qn^2(1 + o(1))$, we conclude that achieving $P_{\mathrm{e}} \to 0$ requires (7).

# C Proof of Theorem 2 (COMP Upper Bound)

Since the random graph is in the typical set (4) with probability approaching one, it suffices to establish that the number of tests (8) yields asymptotically vanishing error probability conditioned on an arbitrary typical graph $G \in \mathcal{T}_n(\epsilon_n)$, with $\epsilon_n = o(1)$ due to Lemma 1. We implicitly condition on such a graph throughout the analysis.

Let $(i, j)$ be a given non-edge of $G$. A particular test fails to identify this non-edge if either (i) $i$ and/or $j$ are not included in the test; or (ii) $i$ and $j$ are both in the test, but there is also an edge covered by the test. Hence, the probability that a given test fails to identify $(i, j)$ as a non-edge is

$$p_0 := (1 - p^2) + p^2\mathbb{P}\big[Y = 1 \,\big|\, \{i, j\} \subseteq \mathcal{L}\big], \tag{33}$$

where we recall that $\mathcal{L}$ is the set of nodes in the test. Note that to obtain $Y = 1$, we need the test to include either a node with an edge connected to $i$ or $j$, or two separate nodes with an edge between them. Denoting these two events by $A_1$ and $A_2$, we have

$$\mathbb{P}\big[Y = 1 \,\big|\, \{i, j\} \subseteq \mathcal{L}\big] \le \mathbb{P}\big[A_1 \,\big|\, \{i, j\} \subseteq \mathcal{L}\big] + \mathbb{P}\big[A_2 \,\big|\, \{i, j\} \subseteq \mathcal{L}\big] \tag{34}$$

$$\le 2dp + \mathbb{P}[Y = 1], \tag{35}$$

where the first term follows because there are at most $2d$ nodes connected to $i$ or $j$, and the second term uses the fact that $A_2$ is independent of the event $\{i, j\} \subseteq \mathcal{L}$ and in itself implies $Y = 1$. Substituting $\mathbb{P}[Y = 1] = (1 - e^\nu)(1 + o(1))$ in accordance with (4), recalling from (6) that $2dp = o(1)$, and returning to (33), we obtain

$$p_0 \le 1 - p^2 + p^2\big((1 - e^\nu)(1 + o(1)) + o(1)\big) \tag{36}$$

$$= 1 - p^2e^{-\nu}(1 + o(1)), \tag{37}$$

since $\nu$ is constant. Hence, the probability that *all* $t$ tests fail to identify $(i, j)$ as a non-edge is

$$p_0^t = \left(1 - p^2 e^{-\nu}(1 + o(1))\right)^t \tag{38}$$

$$\leq e^{-tp^2 e^{-\nu}(1+o(1))}, \tag{39}$$

since $1 - \alpha \leq e^{-\alpha}$. Substituting $p^2 = \frac{\nu}{k}(1 + o(1))$ and setting $\nu = 1$, we obtain

$$p_0^t \leq e^{-\frac{t}{ek}(1+o(1))}, \tag{40}$$

and by a union bound over at most $\binom{n}{2} \leq n^2$ non-edges, it follows that

$$\mathbb{P}[\text{error}] \leq n^2 e^{-\frac{t}{ek}(1+o(1))}. \tag{41}$$

Re-arranging, we deduce that $\mathbb{P}[\text{error}] \to 0$ as long as

$$t \geq \left(2e \cdot k \log n\right)(1 + \eta) \tag{42}$$

for arbitrarily small $\eta > 0$. Since $k = \bar{k}(1 + o(1))$ for all typical graphs, and the probability that $G$ is typical tends to one (see Lemma 1), we obtain the condition in (8).

## D   Proof of Theorem 3 (DD Upper Bound)

Since the random graph is in the typical set (4) with probability approaching one, it suffices to establish that the number of tests (9) yields vanishing error probability conditioned on an arbitrary typical graph $G \in \mathcal{T}_n(\epsilon_n)$, with $\epsilon_n = o(1)$ due to Lemma 1. We implicitly condition on such a graph $G$ throughout the analysis.

### D.1   First Step

The first step of DD gives a set of "possible edges" PE that may contain non-edges. Let $H_0$ be the total number of non-edges in PE, and let $H_1$ be the number of non-edges in PE such that at least one of its two nodes forms part of at least one true edge. Since the total number of non-edges is less than $n^2$, we have from (40) that

$$\mathbb{E}[H_0] \leq n^2 e^{-\frac{t}{ek}(1+o(1))}. \tag{43}$$

Similarly, since the total number of non-edges sharing a node with a true edge is at most $2kd$ (and also trivially less than $n^2$), we have

$$\mathbb{E}[H_1] \leq \min\{2kd, n^2\} e^{-\frac{t}{ek}(1+o(1))}. \tag{44}$$

By Markov's inequality, it follows for any $\xi_0 > 0$ and $\xi_1 > 0$ that that

$$\mathbb{P}[H_0 \geq n^{2\xi_0}] \leq n^{2(1-\xi_0)} e^{-\frac{t}{ek}(1+o(1))} \tag{45}$$

$$\mathbb{P}[H_1 \geq n^{2\xi_1}] \leq \min\{2kd, n^2\} n^{-2\xi_1} e^{-\frac{t}{ek}(1+o(1))}. \tag{46}$$

Re-arranging, we deduce that these two probabilities both vanish as $n \to \infty$ as long as

$$t \geq \left(2(1 - \xi_0)ek \log n\right)(1 + \eta), \tag{47}$$

$$t \geq (1 + \eta)ek \log n \times \begin{cases} 2(1 - \xi_1) & \frac{3}{4} \leq \theta < 1 \\ 4\theta - 1 - 2\xi_1 & \frac{1}{2} < \theta < \frac{3}{4} \\ 2(\theta - \xi_1) & 0 < \theta \leq \frac{1}{2} \end{cases} \tag{48}$$

for arbitrarily small $\eta > 0$; here, the first case uses the $n^2$ term in the $\min\{\cdot\}$ in (46), the second case uses the $2kd$ term and the fact that $k = \Theta(n^{2\theta})$ and $d = \Theta(nq) = \Theta(n^{2\theta-1})$ for $\theta > \frac{1}{2}$, and the third case uses $k = \Theta(n^{2\theta})$ and $d = O(\log n)$ for $\theta \leq \frac{1}{2}$.

It will shortly prove convenient to ensure that $H_0 = o(k)$ and $H_1 = o(\sqrt{k})$ (with high probability). We achieve this by setting $\xi_0$ to be arbitrarily close to (but still less than) $\theta$, and similarly $\xi_1$ arbitrarily close to $\theta/2$, so that the above requirements simplify to

$$t \geq \left(2(1 - \theta)ek \log n\right)(1 + \eta), \tag{49}$$

$$t \geq (1 + \eta)ek \log n \times \begin{cases} 2 - \theta & \frac{3}{4} \leq \theta < 1 \\ 3\theta - 1 & \frac{1}{2} < \theta < \frac{3}{4} \\ \theta & 0 < \theta \leq \frac{1}{2} \end{cases} \tag{50}$$

for arbitrarily small $\eta > 0$.

## D.2 Second Step

We condition on the above-mentioned high-probability events from the first step holding: $H_0 = o(k)$ and $H_1 = o(\sqrt{k})$. In addition, we may assume that the number of positive tests $T_+$ satisfies

$$T_+ = t(1 - e^{-\nu})(1 + o(1)), \tag{51}$$

as this occurs with probability approaching one as $t \to \infty$ in accordance with (28) and standard concentration (e.g., Hoeffding's inequality). We henceforth condition on any such $T_+ = t_+$, as well as a set $\mathrm{PE} = \mathrm{pe}_{h_0,h_1}$ that yields $H_0 = h_0 = o(k)$ and $H_1 = h_1 = o(\sqrt{k})$.

For a given true edge $(i, j)$, let $T_{i,j}$ be the number of tests containing $(i, j)$ and no other edges from PE. We claim that the distribution of $T_{i,j}$ given $t_+$ and $\mathrm{pe}_{h_0,h_1}$ is

$$(T_{i,j} \mid t_+, \mathrm{pe}_{h_0,h_1}) \sim \mathrm{Binomial}\left(t_+, \frac{q_{i,j}}{q_+}\right), \tag{52}$$

where $q_{i,j}$ is the conditional probability (given $\mathrm{PE} = \mathrm{pe}_{h_0,h_1}$) of a given test including $(i, j)$ and no other pairs from PE, and $q_+ = (1 - e^{-\nu})(1 + o(1))$ is the unconditional probability of a positive test. While the distribution (52) is intuitive, its derivation is somewhat tedious, so it is postponed to the end of this appendix (Section D.4).

We proceed by lower bounding $q_{i,j}$. For a given random test, let $A_1$ be the event that the test includes a pair in PE connected to either $i$ or $j$, and let $A_2$ be the event that the test includes a pair in PE connected to neither $i$ nor $j$. Given that $(i, j)$ is in the test (which occurs with probability $p^2$), $(i, j)$ fails to be the unique PE in the test only if either $A_1$ or $A_2$ occurs, so

$$q_{i,j} = p^2 \cdot \left(1 - \mathbb{P}[A_1 \cup A_2 \mid \mathrm{pe}_{h_0,h_1}, \{i, j\} \subseteq \mathcal{L}]\right) \tag{53}$$

$$\geq p^2 \cdot \left(1 - \mathbb{P}[A_1 \mid \mathrm{pe}_{h_0,h_1}, \{i, j\} \subseteq \mathcal{L}] - \mathbb{P}[A_2 \mid \mathrm{pe}_{h_0,h_1}, \{i, j\} \subseteq \mathcal{L}]\right) \tag{54}$$

$$\geq p^2 \cdot \left(1 - (2d + h_1)p - \mathbb{P}[A_2 \mid \mathrm{pe}_{h_0,h_1}, \{i, j\} \subseteq \mathcal{L}]\right) \tag{55}$$

$$= p^2 \cdot \left(1 - o(1) - \mathbb{P}[A_2 \mid \mathrm{pe}_{h_0,h_1}, \{i, j\} \subseteq \mathcal{L}]\right), \tag{56}$$

where the $(2d + h_1)p$ term in (55) arises from at most $2d$ true edges connected to $i$ or $j$ and at most an additional $h_1$ non-edges in PE connected to $i$ or $j$, and (56) follows from the fact that $p = \Theta\left(\frac{1}{\sqrt{k}}\right)$ along with $d = o(\sqrt{k})$ and $h_1 = o(\sqrt{k})$.

To characterize the probability of $A_2$ in (56), we write $A_2 = A_2' \cup A_2''$, where $A_2'$ is the event that the test includes a true edge connected to neither $i$ nor $j$, and $A_2''$ is the event that the test includes a non-edge in PE connected to neither $i$ nor $j$. We have

$$\mathbb{P}[A_2 \mid \mathrm{pe}_{h_0,h_1}, \{i, j\} \subseteq \mathcal{L}] \leq \mathbb{P}[A_2' \mid \mathrm{pe}_{h_0,h_1}, \{i, j\} \subseteq \mathcal{L}] + \mathbb{P}[A_2'' \mid \mathrm{pe}_{h_0,h_1}, \{i, j\} \subseteq \mathcal{L}] \tag{57}$$

$$= \mathbb{P}[A_2'] + \mathbb{P}[A_2'' \mid \mathrm{pe}_{h_0,h_1}, \{i, j\} \subseteq \mathcal{L}] \tag{58}$$

$$\leq \mathbb{P}[Y = 1] + h_0 p^2 \tag{59}$$

$$= (1 - e^{-\nu})(1 + o(1)), \tag{60}$$

where (58) uses the fact that $A_2'$ is independent of all events being conditioned on (since $A_2'$ concerns only true edges separate from $\{i, j\}$), the first term in (59) uses the fact that the event $A_2'$ implies $Y = 1$, the second term in (59) uses the fact that there are at most $h_0$ possible pairs each included with probability $p^2$, and (60) uses $\mathbb{P}[Y = 1] = (1 - e^{-\nu})(1 + o(1))$ along with $h_0 = o(k)$ and $p^2 = \Theta\left(\frac{1}{k}\right)$.

Substituting (60) into (56) gives

$$q_{i,j} \geq p^2 e^{-\nu}(1 + o(1)) \tag{61}$$

$$= \frac{\nu e^{-\nu}}{k}(1 + o(1)), \tag{62}$$

recalling that $p^2 = \frac{\nu}{k}(1 + o(1))$. Returning to (52), we find that

$$T_{i,j} \sim \mathrm{Binomial}\left(t_+, \frac{\nu}{k} \cdot \frac{e^{-\nu}}{1 - e^{-\nu}} \cdot (1 + o(1))\right), \tag{63}$$

and since $t_+ = t(1 - e^{-\nu})(1 + o(1))$ and a $\mathrm{Binomial}(N, r)$ random variable equals zero with probability $(1 - r)^N \le e^{-Nr}$, it follows that

$$\mathbb{P}[T_{i,j} = 0] \le \exp\left( -\frac{t}{k} \cdot \nu e^{-\nu} \cdot (1 + o(1)) \right), \tag{64}$$

and hence

$$\mathbb{P}\left[ \bigcup_{(i,j) \in E} \{T_{i,j} = 0\} \right] \le k \exp\left( -\frac{t}{k} \cdot \nu e^{-\nu} \cdot (1 + o(1)) \right). \tag{65}$$

Re-arranging, setting $\nu = 1$, and writing $\log k = (2\theta \log n)(1 + o(1))$, we find that the second step of DD succeeds as long as

$$t \ge \big(2\theta e k \log n\big)(1 + \eta) \tag{66}$$

for arbitrarily small $\eta > 0$.

### D.3 Combining and Simplifying

To complete the proof of Theorem 3, we only need to show that given the requirements (49) and (66), the additional requirement (50) is redundant (recall also that $k = \bar{k}(1 + o(1))$ for any typical graph $G$). We handle the three cases separately:

- For the first case $\frac{3}{4} \le \theta \le 1$, observe that the coefficient $2 - \theta \le 1.25$ in (50) is strictly less than the coefficient $2\theta \ge 1.5$ in (66).
- For the second case $\frac{1}{2} < \theta < \frac{3}{4}$, observe that the coefficient $3\theta - 1 < 2\theta - 0.25$ in (50) is strictly less than the coefficient $2\theta$ in (66).
- For the third case $0 < \theta \le \frac{1}{2}$, observe that the coefficient $\theta \le \frac{1}{2}$ in (50) is strictly less than the coefficient $2(1 - \theta) \ge 1$ in (49).

### D.4 Derivation of the Conditional Distribution (52)

The derivation of (52) is based on multinomial conditioning, and bears similarity to an analogous conditional distribution for standard group testing [6, Sec. A.3]. To derive the conditional distribution given $t_+$ and $\mathrm{pe}_{h_0,h_1}$, we first need to consider certain unconditional distributions (though still with implicit conditioning on a given typical graph $G$). We define the following random variables:

- $T_-$ is the number of negative tests, $\widetilde{T}_{i,j}$ is the number of tests covering a given true edge $(i, j) \in E$ but no other true edges, and $\widetilde{T}_{\mathrm{extra}}$ is the number of tests covering two or more true edges.
- $T_{i,j}$ is the number of tests covering a given true edge $(i, j) \in E$ and no other pairs from PE.

Since the tests are independent, $(T_-, \{\widetilde{T}_{i,j}\}_{(i,j) \in E}, \widetilde{T}_{\mathrm{extra}})$ has a multinomial distribution with $t$ trials; the corresponding probability parameters are denoted by $(q_-, \{\widetilde{q}_{i,j}\}_{(i,j) \in E}, \widetilde{q}_{\mathrm{extra}})$.

We now consider conditioning on $T_- = t_-$ and $\mathrm{PE} = \mathrm{pe}_{h_0,h_1}$. Under such conditioning, we can characterize the joint distribution of $(\{T_{i,j}\}_{(i,j) \in E}, \{\widetilde{T}_{i,j} - T_{i,j}\}_{(i,j) \in E}, \widetilde{T}_{\mathrm{extra}})$ via the following lemma from [6], stated in generic notation.

**Lemma 2.** [6, Lemma C.1] *Fix the integers $\ell$ and $m$, and let $(W_0, \{W_i\}_{i=1}^\ell, W_{\ell+1})$ have a multinomial distribution with $m$ trials and probabilities $(r_0, \{r_i\}_{i=1}^\ell, r')$. Associate an observation $(W_0, \{W_i\}_{i=1}^\ell, W') = (w_0, \{w_i\}_{i=1}^\ell, w')$ with an unordered list of $m$ class labels (class 0, class $i = 1, \ldots, \ell$, or class $\ell + 1$), and suppose that each label in class $i = 1 \ldots, m$ is independently changed to some class $i'$ with probability $\gamma_i \in [0, 1]$, and to some class $i''$ with probability $1 - \gamma_i$ (where $\gamma_i$ may depend on $w_0$). Then, conditioned on $W_0 = w_0$, the corresponding random variables $(\{W_i'\}_{i=1}^\ell, \{W_i''\}_{i=1}^\ell, W_{\ell+1})$ counting the transformed class labels have a multinomial distribution with $m - w_0$ trials and the following probability parameters:*

$$\left( \left\{ \frac{r_i \gamma_i}{1 - r_0} \right\}_{i=1}^\ell, \left\{ \frac{r_i(1 - \gamma_i)}{1 - r_0} \right\}_{i=1}^\ell, \frac{r'}{1 - r_0} \right). \tag{67}$$

To apply this result, we associate $(T_-, \{\widetilde{T}_{i,j}\}_{(i,j)\in E}, \widetilde{T}_{\text{extra}})$ with $(W_0, \{W_i\}_{i=1}^\ell, W_{\ell+1})$, and associate $(\{T_{i,j}\}_{(i,j)\in E}, \{\widetilde{T}_{i,j} - T_{i,j}\}_{(i,j)\in E}, \widetilde{T}_{\text{extra}})$ with $(\{W_i'\}_{i=1}^\ell, \{W_i''\}_{i=1}^\ell, W_{\ell+1})$. Conditioning on $T_- = t_-$ amounts to conditioning on $W_0$, and conditioning on $\text{PE} = \text{pe}_{h_0,h_1}$ only amounts to changing the value of $\gamma_i$, since PE is determined entirely by the negative tests. Notice that any test contributing (i.e., adding one) to $\widetilde{T}_{i,j}$ further contributes to $T_{i,j}$ independently with probability $\gamma_{i,j}$, defined to be the conditional probability that some non-edge in $\text{pe}_{h_0,h_1}$ is covered by the test given that $(i,j)$ is the unique true edge covered. Hence, given $T_- = t_-$ and $\text{PE} = \text{pe}_{h_0,h_1}$, Lemma 2 implies that the random variables $(\{T_{i,j}\}_{(i,j)\in E}, \{\widetilde{T}_{i,j} - T_{i,j}\}_{(i,j)\in E}, \widetilde{T}_{\text{extra}})$ have a multinomial distribution with $t_+ = t - t_-$ trials and the following probability parameters:

- For $(i,j) \in E$, the parameter for $T_{i,j}$ is $\frac{\widetilde{q}_{i,j}\gamma_{i,j}}{1-q_-}$;
- For $(i,j) \in E$, the parameter for $\widetilde{T}_{i,j} - T_{i,j}$ is $\frac{\widetilde{q}_{i,j}(1-\gamma_{i,j})}{1-q_-}$;
- The parameter for $\widetilde{T}_{\text{extra}}$ is $\frac{\widetilde{q}_{\text{extra}}}{1-q_-}$.

We conclude by showing that (52) follows from the first of these dot points, with the marginal distribution of a multinomial distribution being binomial. The denominator $1 - q_-$ is trivially equal to $q_+$, and the numerator $\widetilde{q}_{i,j}\gamma_{i,j}$ equals the product of two terms. To understand these terms, let $\widetilde{B}_{i,j}$ be the event that a given test covers $(i,j)$ but no other true edge, and let $B_{i,j}$ be the event that it covers $(i,j)$ but no other pair from PE. Then, the previous definitions can be written as

$$\widetilde{q}_{i,j} = \mathbb{P}[\widetilde{B}_{i,j}], \quad \gamma_{i,j} = \mathbb{P}[B_{i,j} \,|\, \text{pe}_{h_0,h_1}, \widetilde{B}_{i,j}]. \tag{68}$$

In addition, we have $\mathbb{P}[\widetilde{B}_{i,j}] = \mathbb{P}[\widetilde{B}_{i,j} \,|\, \text{pe}_{h_0,h_1}]$, since $\widetilde{B}_{i,j}$ is independent of PE (note that PE is determined entirely by the negative tests, and $\widetilde{B}_{i,j}$ only concerns true edges). As a result, we have

$$\widetilde{q}_{i,j}\gamma_{i,j} = \mathbb{P}[\widetilde{B}_{i,j} \,|\, \text{pe}_{h_0,h_1}]\mathbb{P}[B_{i,j} \,|\, \text{pe}_{h_0,h_1}, \widetilde{B}_{i,j}] \tag{69}$$

$$= \mathbb{P}[\widetilde{B}_{i,j} \cap B_{i,j} \,|\, \text{pe}_{h_0,h_1}] \tag{70}$$

$$= \mathbb{P}[B_{i,j} \,|\, \text{pe}_{h_0,h_1}], \tag{71}$$

where (71) follows since $B_{i,j}$ implies $\widetilde{B}_{i,j}$, because all true edges are in PE with probability one (i.e., the first step of DD has no false negatives). Finally, (71) coincides precisely with the definition of $q_{i,j}$ stated following (52), and this completes the derivation of (52).

## E Proof of Theorem 4 (SSS Lower Bound)

Since the random graph is in the typical set (4) with probability approaching one, it suffices to establish that the number of tests (10) yields error probability tending to one conditioned on an arbitrary typical graph $G \in \mathcal{T}_n(\epsilon_n)$, with $\epsilon_n = o(1)$ due to Lemma 1. We implicitly condition on such a graph $G$ throughout the analysis.

Let $M_{ij}$ be the event that edge $(i,j)$ is *masked*, i.e., whenever its nodes both appear in a test, the nodes of some different edge are also included in the test. In this case, there exists a satisfying set (of edges) of cardinality $k - 1$, so the algorithm will fail to output the true edge set. Hence,

$$P_{\text{e}} \geq \mathbb{P}\left[\bigcup_{(i,j)\in E} M_{ij}\right] \tag{72}$$

$$\geq \sum_{(i,j)\in E} \frac{\mathbb{P}[M_{ij}]^2}{\sum_{(i',j')\in E} \mathbb{P}[M_{ij} \cap M_{i'j'}]}, \tag{73}$$

where (73) is an application of de Caen's bound [22].

We proceed by bounding the individual and pairwise masking probabilities. For a given edge $(i,j)$ to be masked, for each of the $t$ tests we need either $i$ or $j$ to be excluded, or for the nodes of some other edge to be included. Letting $A_1^{(ij)}$ be the event that some other node connected to $i$ or $j$ is included, and $A_2^{(ij)}$ the event that two connected nodes distinct from $i$ and $j$ are included, the associated masking event for a single test has probability

$$p_1^{(ij)} = (1 - p^2) + p^2\mathbb{P}[A_1^{(ij)} \cup A_2^{(ij)} \,|\, \{i,j\} \subseteq \mathcal{L}]. \tag{74}$$

We lower bound $p_1^{(ij)}$ by ignoring the event $A_1^{(ij)}$:

$$p_1^{(ij)} \geq (1 - p^2) + p^2 \mathbb{P}[A_2^{(ij)} \,|\, \{i,j\} \subseteq \mathcal{L}] \tag{75}$$

$$= 1 - p^2 + p^2 \mathbb{P}[A_2^{(ij)}], \tag{76}$$

since $A_2^{(ij)}$ is independent of whether $\{i,j\} \subseteq \mathcal{L}$. Now observe that the unconditional probability of a positive test satisfies

$$\mathbb{P}[Y = 1] = \mathbb{P}\big[\{\{i,j\} \subseteq \mathcal{L}\} \cup A_1^{(ij)} \cup A_2^{(ij)}\big] \tag{77}$$

$$\leq \mathbb{P}\big[\{i,j\} \subseteq \mathcal{L}\big] + \mathbb{P}\big[A_1^{(ij)}\big] + \mathbb{P}\big[A_2^{(ij)}\big] \tag{78}$$

$$\leq p^2 + 2dp^2 + \mathbb{P}\big[A_2^{(ij)}\big], \tag{79}$$

and hence

$$\mathbb{P}\big[A_2^{(ij)}\big] \geq P_Y(1) - \xi, \tag{80}$$

where $P_Y(1)$ is a shorthand for $\mathbb{P}[Y = 1]$, and $\xi = (1 + 2d)p^2$. Substitution into (76) gives

$$p_1^{(ij)} \geq 1 - p^2(1 - P_Y(1) + \xi). \tag{81}$$

Next, we upper bound $p_1^{(ij)}$. Applying the union bound in (74), we obtain

$$p_1^{(ij)} \leq (1 - p^2) + p^2\big(\mathbb{P}[A_1^{(ij)} \,|\, \{i,j\} \subseteq \mathcal{L}] + \mathbb{P}[A_2^{(ij)} \,|\, \{i,j\} \subseteq \mathcal{L}]\big) \tag{82}$$

$$\leq (1 - p^2) + p^2\big(2dp + \mathbb{P}[A_2^{(ij)}]\big) \tag{83}$$

$$\leq 1 - p^2\big(1 - P_Y(1) - \xi'\big) \tag{84}$$

where (83) uses $\mathbb{P}[A_1^{(ij)} \,|\, \{i,j\} \subseteq \mathcal{L}] \leq 2dp$ and the fact that $A_2^{(ij)}$ is independent of whether $\{i,j\} \subseteq \mathcal{L}$, and (84) uses $\mathbb{P}[A_2^{(ij)}] \leq P_Y(1)$ (see (77)) and defines $\xi' = 2dp$.

Now, for the masking event $M_{ij}$ to occur, the probability-$p_1^{(ij)}$ masking event needs to occur for all tests, yielding $\mathbb{P}[M_{ij}] = (p_1^{(ij)})^t$. Moreover, for both $M_{ij}$ and $M_{i'j'}$ to occur, the case $(i,j) = (i',j')$ is handled trivially, whereas for $(i,j) \neq (i',j')$ the associated events for $(i,j)$ and $(i',j')$ need to occur simultaneously for each test. Since the complementary event (i.e., the edge is the only one covered by the nodes included in the test) can only occur for one of $(i,j)$ or $(i',j')$, the associated probability $p_1^{(ij \cap i'j')}$ of both masking events occurring for a single test satisfies

$$1 - p_1^{(ij \cap i'j')} = (1 - p_1^{(ij)}) + (1 - p_1^{(i'j')}), \tag{85}$$

i.e., $\mathbb{P}[A \cup B] = \mathbb{P}[A] + \mathbb{P}[B]$ for disjoint events $A$ and $B$. Hence, from (84),

$$p_1^{(ij \cap i'j')} \leq 1 - 2p^2\big(1 - P_Y(1) - \xi'\big). \tag{86}$$

Taking the intersection over the $t$ tests gives $\mathbb{P}[M_{ij} \cap M_{i'j'}] = (p_1^{(ij \cap i'j')})^t$ for all $(i,j) \neq (i',j')$, and substituting the preceding findings into (73) gives

$$\mathbb{P}[\text{error}] \geq \sum_{(i,j)\in E} \frac{\big(p_1^{ij}\big)^{2t}}{\big(p_1^{ij}\big)^t + \sum_{(i',j')\neq(i,j)} \big(p_1^{(ij \cap i'j')}\big)^t} \tag{87}$$

$$\geq \sum_{(i,j)\in E} \frac{\big(1 - p^2(1 - P_Y(1) + \xi)\big)^{2t}}{\big(1 - p^2(1 - P_Y(1) - \xi')\big)^t + \sum_{(i',j')\neq(i,j)} \big(1 - 2p^2(1 - P_Y(1) - \xi')\big)^t} \tag{88}$$

$$\geq \frac{k\big(1 - p^2(1 - P_Y(1) + \xi)\big)^{2t}}{\big(1 - p^2(1 - P_Y(1) - \xi')\big)^t + k\big(1 - 2p^2(1 - P_Y(1) - \xi')\big)^t}, \tag{89}$$

since $|E| = k$ (in the denominator, we upper bound $k - 1 \leq k$).

We upper bound the terms in the denominator in (89) using $1 - \alpha \le e^{-\alpha}$, and characterize the numerator using $1 - \alpha = e^{-\alpha + O(\alpha^2)}$ as $\alpha \to 0$ (recall that $p^2 = \Theta\left(\frac{1}{k}\right) = o(1)$ and $P_Y(1) = \Theta(1)$):

$$\mathbb{P}[\text{error}] \ge \frac{ke^{-2t\left(p^2(1-P_Y(1)+\xi)+O(p^4)\right)}}{e^{-tp^2(1-P_Y(1)-\xi')} + ke^{-2tp^2(1-P_Y(1)-\xi')}} \tag{90}$$

$$\ge \frac{ke^{-tp^2(1-P_Y(1))}}{1+ke^{-tp^2(1-P_Y(1))}} \cdot \frac{e^{-2t(p^2\xi+O(p^4))}}{e^{2tp^2\xi'}}. \tag{91}$$

Since the converse bound we are proving is of the form $t = \Omega(k \log n)$, we can assume without loss of generality that $t = \Theta(k \log n)$, as additional tests can only help the SSS algorithm.[7] In addition, we can assume without loss of generality that $p^2 = \Theta\left(\frac{1}{k}\right)$, since if $p^2$ behaves as $o\left(\frac{1}{k}\right)$ or $\omega\left(\frac{1}{k}\right)$ then the probability of a positive test tends to 0 or 1 as $n \to \infty$, and it follows from a standard entropy-based argument that $\omega(k \log n)$ tests are needed [7, Lemma 1]. We claim that these conditions imply that

$$\frac{e^{-2t(p^2\xi+O(p^4))}}{e^{2tp^2\xi'}} \to 1. \tag{92}$$

This is seen by noting that $tp^2 = \Theta(\log n)$ by the above-mentioned behavior of $t$ and $p^2$, whereas the terms $\xi = (1+2d)p^2$, $\xi' = 2dp$, and $O(tp^4)$ all behave as $O(n^{-c})$ for sufficiently small $c$. This behavior is easy to see for the $O(tp^4)$ term by the above-mentioned behavior of $t$ and $p^2$, and is seen to also hold for $\xi$ and $\xi'$ by noting that $dp = \Theta\left(\frac{d}{\sqrt{k}}\right)$, along with $d = O(\max\{\log n, nq\})$ (see (5)), $\sqrt{k} = \Theta(n\sqrt{q})$, and the behavior of $q$ in (3).

Substituting (92) into (91), we have

$$\mathbb{P}[\text{error}] \ge \frac{1}{1+ke^{tp^2(1-P_Y(1))}}(1+o(1)), \tag{93}$$

and substituting $p^2 = \frac{\nu}{k}$ and $1 - P_Y(1) = e^{-\nu}(1 + o(1))$, we deduce that $\mathbb{P}[\text{error}] \to 1$ whenever

$$t \le \frac{k \log k}{\nu e^{-\nu}}(1-\eta) \tag{94}$$

for arbitrarily small $\eta > 0$. Since the function $\nu e^{-\nu}$ is maximized at $\nu = 1$, we deduce that $\mathbb{P}[\text{error}] \to 1$ whenever

$$t \le \left(ke \log k\right)(1-\eta). \tag{95}$$

The proof is completed by recalling that for any typical graph, $k = \bar{k}(1 + o(1))$ and $\log k = (2\theta \log n)(1 + o(1))$ (since $\bar{k} = \Theta(n^{2\theta})$).

## F  Missing Details in the Proof of Theorem 5 (Sublinear-Time Decoding)

### F.1  Details of Step 1 – Bundles of Tests

Recall that we form a number $B$ of "bundles" of tests, where each node is placed in each bundle with probability $r \in (0,1)$. For a given bundle, consider the probability $p_{\text{one}}$ of a given edge $(i,j)$ being the only edge among its nodes. Letting $A_1$ be the event that some other node connected to $i$ or $j$ is in the bundle, and letting $A_2$ be the event that two different edge-connected nodes are in the bundle, we have

$$p_{\text{one}} = r^2 \cdot \mathbb{P}[A_1^c \cap A_2^c] \tag{96}$$

$$\ge r^2 \cdot \left(1 - \mathbb{P}[A_1] - \mathbb{P}[A_2]\right) \tag{97}$$

$$\ge r^2 \cdot \left(1 - 2dr - kr^2\right), \tag{98}$$

where (98) uses the fact that there are at most $2d$ nodes connected to $i$ or $j$, and at most $k$ other edges separate from $i$ and $j$. Setting $r = \frac{1}{\sqrt{2\bar{k}}}$ gives

$$p_{\text{one}} \ge \frac{1}{2\bar{k}}\left(1 - \frac{2d}{\sqrt{2\bar{k}}} - \frac{1}{2} \cdot \frac{k}{\bar{k}}\right) \tag{99}$$

$$= \frac{1}{4\bar{k}}(1+o(1)), \tag{100}$$

Figure 3: (Left) Number of tests for COMP, DD, and LP under four different $(n, \overline{k})$ pairs: $n \in \{80, 100, 120, 140\}$ and $\overline{k} = \frac{n}{10}$. (Right) Normalized number of tests after division by $\overline{k} \log \frac{1}{q}$, where $q$ is the probability of each edge in the graph.

since $d \ll \sqrt{k}$ (see (6)) and $k = \overline{k}(1 + o(1))$. Hence, the probability of $(i, j)$ being the unique edge in *some* bundle satisfies

$$p_{\text{any}} = 1 - \left( 1 - \frac{1}{4\overline{k}}(1 + o(1)) \right)^B$$

$$\geq 1 - e^{-\frac{B}{4\overline{k}}(1 + o(1))},$$

and by a union bound over the $k = \overline{k}(1 + o(1))$ edges, we find that $B = \left( 4\overline{k} \log \overline{k} \right)(1 + o(1))$ bundles suffice to ensure that every edge is the unique one in at least one bundle.

### F.2   Details of Step 4 – Total Number of Tests and Decoding Time

The number of tests used is asymptotically dominated by that of the location tests, and recalling that $B = \left( 4\overline{k} \log \overline{k} \right)(1 + o(1))$, we find that $t = 4\overline{k}(\log \overline{k})(\log_2 n)^2(1 + o(1))$. We briefly mention that this can be significantly reduced when adaptivity is allowed, similarly to standard group testing [18], but our focus in this paper is on the non-adaptive setting.

For the decoding time, we notice that each multiplicity test takes $O(\log B)$ time (i.e., the same as the number of tests used), whereas for the location test we can actually make the decoding time less than the number of tests due to the fact that we don't end up making use of most test outcomes.[8] Specifically, each iteration from $\ell = 1, \ldots, L$ observes at most 3 test outcomes and runs in $O(1)$ time, so the overall time per location test is $O(L)$. Since the decoder makes use of $B$ multiplicity tests and $k$ location tests (assuming no errors occur), the total runtime is $O(B \log B + O(kL))$, which simplifies to $O(\overline{k} \log^2 \overline{k} + \overline{k} \log n)$.

## G   Additional Numerical Experiments

In order to demonstrate that the empirical performance of our algorithms is in agreement with our theory, we plot the success probability as a function of the number of tests for various $(\overline{k}, n)$ pairs, and then re-plot them with the number of tests normalized by $\overline{k} \log \frac{1}{q}$ (e.g., see Theorem 1; similar normalization is also used in Figure 1). The results, averaged over 1000 trials, are shown in Figure 3.

As predicted by our theory, the resulting curves for each algorithm are in general agreement after performing the normalization, with slight deviations due to noise and non-asymptotic considerations. Moreover, according to the sparse regime of Figure 1, our theory suggests (for sufficiently sparse settings) an asymptotic threshold of roughly 1 for the optimal algorithm (which LP approximates), roughly 2 for DD, and slightly over 2 for COMP. The above figure is consistent with these numbers, though they are slightly increased because of the penalty incurred for finite $n$ (as opposed to $n \to \infty$), and possibly also the choice $\overline{k} = \frac{n}{10}$ (as opposed to $\overline{k} \sim n^{2\theta}$).

## H    Results for General Edge and Degree Bounded Graphs

Since the assumption of independent edges is not always appropriate for modeling real-world applications, there is substantial motivation to develop performance bounds that hold with high probability for any given graph in a *deterministic graph class*.

An impossibility result of [1] shows that if one only fixes the number of edges to $k$, then achieving $t = O(k \log n)$ scaling is not possible in the worst case. A natural question is then whether fixing the number of edges $k$ *and* maximum degree $d$ results in $t = O(k \log n)$ scaling under suitable assumptions on $d$. In this section, we argue that the answer is affirmative as long as $d = o(\sqrt{k})$.

Indeed, an inspection of our analysis reveals that the condition $d \leq d_{\max}$ in the typical set (4) was not used directly, but rather, was only used to establish (6). On the other hand, the condition $\mathbb{P}_G[Y = 1] = (1 - e^{-\nu})(1 + o(1))$ played a significant role in our analysis, and it is unclear whether it can be deduced from the condition $d = o(\sqrt{k})$ alone.

However, while exactly characterizing $\mathbb{P}_G[Y = 1]$ for an arbitrary edge-bounded and degree-bounded graph $G$ may be difficult, we can easily find upper and lower bounds. First, by the union bound, we have

$$\mathbb{P}_G[Y = 1] \leq kp^2 = \nu, \tag{101}$$

under the choice $p = \sqrt{\frac{\nu}{k}}$. As for the lower bound, applying de Caen's bound [22] and letting $A_1, \ldots, A_k$ be the events of the $k$ edges having both their nodes included in the test, we have

$$\mathbb{P}_G[Y = 1] = \mathbb{P}_G\left[ \bigcup_{i=1,\ldots,k} \{A_i\} \right] \tag{102}$$

$$\geq \sum_{i=1,\ldots,k} \frac{\mathbb{P}_G[A_i]^2}{\mathbb{P}_G[A_i] + \sum_{j \neq i} \mathbb{P}_G[A_i \cap A_j]}. \tag{103}$$

Note that $\mathbb{P}_G[A_i] = p^2$ for all $i$, since the two nodes of the edge need to be included.

Among the terms $\sum_{j \neq i} \mathbb{P}_G[A_i \cap A_j]$, there are at most $2d$ terms for which the two associated edges share a node ($d$ per node times two nodes), and for those terms we have $\mathbb{P}_G[A_i \cap A_j] = p^3$. All other terms (of which there are at most $k$) have $\mathbb{P}_G[A_i \cap A_j] = p^4$, and hence

$$\mathbb{P}_G[Y = 1] \geq \frac{kp^4}{p^2 + 2dp^3 + kp^4} \tag{104}$$

$$= \frac{\nu}{1 + \nu}(1 + o(1)), \tag{105}$$

where we have used $p = \sqrt{\frac{\nu}{k}}$ and $dp \ll 1$.

With the above upper and lower bounds on $\mathbb{P}_G[Y = 1]$ in place, upon fixing $\nu \in (0, 1)$ (e.g., $\nu = \frac{1}{2}$), the rest of the analysis of COMP, DD, and SSS proceeds similarly to that done for the graphs in the typical set (4),[9] and yields analogous results with $O(k \log n)$ scaling, albeit slightly worse constant factors. For GROTESQUE, the extension is even more immediate, since we did not use any characteristics of $\mathbb{P}_G[Y = 1]$ in its analysis.

## I    Challenges in the Analysis Compared to Standard Group Testing

Recall that the standard group testing problem concerns recovering a defective set $S \subseteq \{1, \ldots, N\}$ of cardinality $K$ from a set of $N$ items, using a sequence of tests in groups of items [8, 23]. Each test returns one if there is at least one defective item in the test, and zero otherwise. In this section, we highlight some of the main challenges arising in our analysis of COMP, DD, SSS, and GROTESQUE compared to their counterparts for standard group testing [6, 18, 19].

An immediate challenge is that in contrast with group testing, the analysis is not symmetric with respect to graphs having a given number of edges (e.g., the degree of each node also plays a major role). Related to this issue is the fact that the events associated with including two different edges in

a given test are *not* independent if those edges have a node in common. As a a result, we frequently need to distinguish between error events for neighbors vs. non-neighbors of a given node pair $(i, j)$.

For the COMP algorithm (*cf.*, Section 4.1), our analysis closely follows that of group testing after Lemma 1 (establishing the asymptotic behavior of $\mathbb{P}_G[Y = 1]$) is established. However, as seen in Appendix A, the proof of that lemma was in itself highly non-trivial.

For DD (*cf.*, Section 4.2), in addition to Lemma 1, additional effort is needed to handle the fact that different graphs lead to different probabilities of the key error events, which is not the case in group testing. The first step (Section D.1), distinguishes between the total number of non-edges in PE, and the number that actually share a node in common with a true edge, leading to several conditions on $t$ in (49)–(50) that luckily end up simplifying in Section D.3. This also impacts the analysis of the second step (Section D.2), where we require a careful analysis in (53)–(60) to characterize the key "success event" of including a given edge and no other pairs from the set PE.

For SSS, (*cf.*, Section 5), we face similar difficulties in bounding the individual and pairwise events in (73), with a more delicate analysis leading to the remainder terms $\xi$ and $\xi'$ in (80) and (84). To complete the analysis after (91), it is essential that these remainder terms not only behave as $o(1)$, but decay to zero *sufficiently fast*.

For GROTESQUE (*cf.*, Section 6), the multiplicity test (*cf.*, Section 6.2) is a fairly straightforward extension of that of standard group testing. However, the location test (*cf.*, Section 6.3) is more novel. In group testing, each item can be assigned a unique length-$L$ binary string, and the single defective item under consideration can trivially be identified using one test per bit. In our setting, we need to design tests that simultaneously identify two nodes, and when the two bits of the corresponding strings differ, we need to ensure that the bit assignments are done in a consistent manner across the $L$ indices (*cf.*, Step 3(b) in Section 6.3).

## Footnotes

[6] For some combinations of $j$ and $\ell$ we trivially have $U_{j,\ell}(G) = 0$, but there is no need for us to explicitly account for this.

[7] If an incorrect set is satisfying with respect to a certain number of tests, it remains satisfying after removing any subset of those tests. Hence, removing tests cannot decrease the error probability.

[8]We still need to perform such tests, because *a priori* we don't know which $O(L)$ size subset of the $O(L^2)$ relevant tests performed will end up being useful.

[9]For SSS, we need to slightly strengthen the assumption $d = o(\sqrt{k})$ to $d = o\left(\frac{\sqrt{k}}{\log n}\right)$; see the part of the proof following (92).