[Reviews · NeurIPS 2019]

Reviewer 1



The paper is well written and answers some interesting theoretical questions at the intersection of non-adaptive group testing and graph learning. The problem, thought connected to structured group testing, seems novel in this particular incarnation. While the algorithms proposed are not especially novel, the authors needed too perform some nuanced analysis, and I particularly appreciate Appendix H that outlines the difficulties. Overall, this is a strong paper with interesting results. I recommend its acceptance.

Reviewer 2



This kind of nonadaptive testing for graph recovery problem seems to arise in numerous settings and is therefore natural and important; the sparse Erdos-Renyi setting is perhaps the most basic for average-case analysis and so the focus there makes sense (there is a brief discussion of general graphs). Although I didn't check each proof, line-by-line, the high-level ideas of everything indicate correctness to me. The fact the simulations qualitatively line up is also suggestive. There is a lot of content in this paper, with several distinct achievable schemes (4 by my count), of which one is near-optimal, whereas the others have potentially interesting relationships among them. Although the writing is generally well-organized and clear, I wonder if so much material is perhaps ambitious for an 8-page conference paper. In particular, a significant part of the content, including pretty much all proofs are deferred to appendices. Although 1-sentence sketch of proofs are often given in text, a little more detail in the main text would have been nice, especially about the sublinear time decoding algorithm. Moreover, Appendix H on comparing standard group testing to this setting would have been appreciated in the main text, to help clarify novelty. I should say, there is certainly novelty. ADDENDUM: I have read through the other reviewer comments as well as the author rebuttal, and my scoring remains the same based on the theoretical contribution. I appreciate the authors will try to include more content by more concise formatting and the extra page; based on R4's comments as representative of broader readership, perhaps one should ignore my previous suggestion of reducing the scope of the experimental section. R4's suggestion of releasing code is also a good one, for reproducibility and greater usage of your algorithm.

Reviewer 3



This paper studies a learning task of Erdos-Renyi graph with edge detecting queries. Each query or test is given as a set of input (a set of nodes) X representing a set of nodes and output Y(X)=0 or 1 representing the absence or presence of at lease one edge in the node set X. The authors derive the theoretical and algorithmic bounds of the number of tests required for achieving the perfect reconstruction of the graph in the asymptotic limit where the number of nodes $n$ tends to be infinity, using three known algorithms: COMP, DD, and SSS. The derived analytic form is tight up to the asymptotic constant. A new algorithm is also invented based on GROTESQUE algorithm, achieving a sublinear computational time in the decoding step which is near optimal compared to the derived bounds. Originality: The task is new and the novelty of the contribution is clear. Quality: The submission is technically sound. All claims are proved by using standard bounding techniques (Chernoff, union) plus some more advanced techniques. Both the strong and weak points of the proposed methods are honestly pointed out. In this sense, the quality is high. Clarity: The paper is basically well organized, though I sometimes found some uneasy expressions. For example, at first sight, it is not easy to understand Figure 1, because the meaning of some parameters (e.g. \theta) and of the axes are not well explained up to the point where Figure 1 is first referred in the main text. Below I give the list of typos and uneasy expressions which I found: Line 4: hard the sense -> hard in the sense Line 22-24: The sentence Each node is ... in that query.'' is hard to understand. Line 663: then then -> then Significance: Both the derived bounds and the proposed algorithm are interesting and can be useful also for practical purposes. The significance thus is high. Additional comments after feedback: Given the author feedback and discussions with the other reviewers, I am convinced with the importance of this work. The score is increased accordingly. But still I think numerical experiments and codes are valuable, because for practical users they are more convincing and important.

[Author Response · NeurIPS 2019]

# Author Responses for "Learning Erdős-Rényi Random Graphs via Edge Detecting Queries"

We are very pleased to have received these positive comments on our paper, and we are grateful to all of the reviewers for their feedback and suggestions.

**Response to Reviewer 3:** By freeing up some space as suggested, as well as making use of the 9th page allowed in the camera-ready version, we can include the details of the sublinear-time decoding algorithm in the main body. We will also elaborate on as many proof outlines as possible.

**Response to Reviewer 4:** We are grateful for the feedback, but are admittedly a little surprised at the final score entered, given the reviewer's positive comments on all four of Originality, Quality, Clarity, and Significance.

Regarding the minor clarity issues, we will adjust Figure 1 according to these suggestions and fix the typos stated.

If we understand correctly, the reviewer's main concerns are that the numerical results are not comprehensive. We appreciate this type of concern in general, but we would like to emphasize that this is a theory paper, and that the experiments are only meant to serve as a simple proof-of-concept rather than being central to the paper. (This is supported by Reviewer 3's suggestion to in fact make the experiment section *more compact*.) Please also note that the majority of related works on this problem and group testing (see our References section) do not include experiments. In light of all this, we hope that the final decision is made based on the theory.

We compared COMP/DD/SSS/LP experimentally because these all use the same test matrix (i.i.d. Bernoulli) so give more directly comparable results. GROTESQUE is not meant to compete with these in terms of the number of tests, as a factor $O(\log \overline{k} \cdot \log n)$ increase is typically significant in practice. (We would be happy to highlight this further in Section 6). Again, we believe that adopting the current format is appropriate for a theory paper, and that improved sublinear-time decoding algorithms (e.g., optimal scaling laws, improved constants, and/or competitive empirical performance) are better left for future work, especially given that the number of results and their level of detail is already quite high for a conference paper.

Comparing the empirical number of tests to the theoretical thresholds is a good idea, though in accordance with Reviewer 3's suggestions, we believe it would belong in the supplementary material and not the main body. The following figure illustrates an example for COMP (green / right), DD (blue / middle), and LP (red / left):

Figure 1: (Left) Number of tests for four different $(n, \overline{k})$ pairs described below; (Right) Normalized number of tests after division by $\overline{k} \log \frac{1}{q}$, where $q$ is the probability of each edge in the graph.

Here, we have run the algorithms with $n \in \{80, 100, 120, 140\}$ and $\overline{k} = \frac{n}{10}$, and normalized the horizontal axis of the second plot to $\#\text{tests}/(\overline{k} \log \frac{1}{q})$. As predicted by our theory, the resulting 4 curves for each algorithm nearly overlap, with slight deviations due to noise and a sharper threshold as $n$ increases. Moreover, the theory (see Figure 1 of the paper with a sparsity level close to zero) suggests an asymptotic threshold of roughly 1 for the optimal algorithm (which LP approximates), roughly 2 for DD, and slightly over 2 for COMP. The above figure is consistent with these numbers, though they are slightly increased because of the penalty incurred for finite $n$ (as opposed to $n \to \infty$).

[Meta-Review · NeurIPS 2019]

This paper studies average-case performance of learning Erdos-Renyi random graphs via non-adaptive edge-detecting queries. The main contributions are threefold: - An algorithm-independent probabilistic lower bound of the number of non-adaptive tests is given via Fano's inequality (Theorem 1). - Three existing algorithms for standard group testing, COMP, DD, and SSS, are extended to the graph learning problem. Under the Bernoulli testing, upper bounds are provided for COMP (Theorem 2) and DD (Theorem 3), and a lower bound is provided for SSS (Theorem 4). These in particular show that DD is asymptotically optimal under the Bernoulli testing when \theta>1/2. - A sublinear-time algorithm based on the GROTESQUE algorithm for group testing is proposed, and bounds on the number of tests and on decoding time are given for that algorithm (Theorem 5). Although the three review scores exhibited a relatively large split in the initial round of review, after the authors' rebuttal as well as discussion among the reviewers, all the three reviewers have become positive ratings. I would thus like to recommend acceptance of this paper. Minor point: Line 204: unless one moves (bound -> beyond?) Bernoulli test designs.